# Dealing With Misspecification In Fixed-Confidence Linear Top-m Identification

**Clémence Réda**
Université de Paris, NeuroDiderot, Inserm, F-75019 Paris, France
clemence.reda@inria.fr

**Andrea Tirinzoni**
Univ. Lille, Inria, CNRS, Centrale Lille, UMR 9189 CRIStAL, F-59000 Lille, France
andrea.tirinzoni@inria.fr

**Rémy Degenne**
Univ. Lille, Inria, CNRS, Centrale Lille, UMR 9189 CRIStAL, F-59000 Lille, France
remy.degenne@inria.fr

## Abstract

We study the problem of the identification of $m$ arms with largest means under a fixed error rate $\delta$ (fixed-confidence Top-$m$ identification), for misspecified linear bandit models. This problem is motivated by practical applications, especially in medicine and recommendation systems, where linear models are popular due to their simplicity and the existence of efficient algorithms, but in which data inevitably deviates from linearity. In this work, we first derive a tractable lower bound on the sample complexity of any $\delta$-correct algorithm for the general Top-$m$ identification problem. We show that knowing the scale of the deviation from linearity is necessary to exploit the structure of the problem. We then describe the first algorithm for this setting, which is both practical and adapts to the amount of misspecification. We derive an upper bound to its sample complexity which confirms this adaptivity and that matches the lower bound when $\delta \to 0$. Finally, we evaluate our algorithm on both synthetic and real-world data, showing competitive performance with respect to existing baselines.

## 1 Introduction

The multi-armed bandit (MAB) is a popular framework to model sequential decision making problems. At each round $t > 0$, a learner chooses an *arm* $k_t$ among a finite set of $K \in \mathbb{N}$ possible options, and it receives a random reward $X_t^{k_t} \in \mathbb{R}$ drawn from a distribution $\nu^{k_t}$ with unknown mean $\mu^{k_t}$. Among the many problem settings studied in this context, we focus on *pure exploration*, where the learner aims at maximizing the information gain for answering a given query about the arms [5]. In particular, we are interested in finding a subset of $m \geq 1$ arms with largest expected reward, which is known as the *Top-$m$ identification* problem [22]. This generalizes the widely-studied best-arm (i.e., Top-1) identification problem [16]. This problem has several important applications, including online recommendation and drug repurposing [31, 35]. Two objectives are typically studied. On the one hand, in the *fixed-budget* setting [2], the learner is given a finite amount of samples and must return a subset of $m$ best arms while minimizing the probability of error in identification. On the other hand, in the *fixed-confidence* setting [16], the learner aims at minimizing the *sample complexity* for returning a subset of $m$ best arms with a fixed maximum error rate $\delta \in (0, 1)$, defined as the number of samples collected before the algorithm stops. This paper focuses on the latter.

35th Conference on Neural Information Processing Systems (NeurIPS 2021).

In practice, information about the arms is typically available (e.g., the characteristics of an item in a recommendation system, or the influence of a drug on protein production in a clinical application). This side information influence the expected rewards of the arms, thus adding *structure* (*i.e.*, prior knowledge) to the problem. This is in contrast to the classic *unstructured* MAB setting, where the learner has no prior knowledge about the arms. Due to their simplicity and flexibility, linear models have become the most popular to represent this structure. Formally, in the *linear bandit* setting [3], the mean reward $\mu^k$ of each arm $k \in [K] := \{1, 2, \dots, K\}$ is assumed to be an inner product between known $d$-dimensional arm features $\phi_k \in \mathbb{R}^d$ and an unknown parameter $\theta \in \mathbb{R}^d$. This model has led to many provably-efficient algorithms for both best-arm [38, 42, 17, 43, 13] and Top-$m$ identification [24, 35]. Unfortunately, the strong guarantees provided by these algorithms hold only when the expected rewards are perfectly linear in the given features, a property that is often violated in real-world applications. In fact, when using linear models with real data, one inevitably faces the problem of *misspecification*, i.e., the situation in which the data deviates from linearity.

A *misspecified linear bandit* model is often described as a linear bandit model with an additive term to encode deviation from linearity. Formally, the expected reward $\mu^k = \phi_k^\top \theta + \eta^k$ of each arm $k \in [K]$ can be decomposed into its linear part $\phi_k^\top \theta$ and its misspecification $\eta^k \in \mathbb{R}$. Note the flexibility of this model: for $\|\eta\| = 0$, where $\eta = [\eta^1, \eta^2, \dots, \eta^K]^\top$, the problem is perfectly linear and thus highly structured, as the mean rewards of different arms are related through the common parameter $\theta$; whereas when the misspecification vector $\eta$ is large in all components, the problem reduces to an unstructured one, since knowing the linear part alone provides almost no information about the expected rewards. Learning in this setting thus requires adapting to the scale of misspecification, typically under the assumption that some information about the latter is known (e.g., an upper bound $\varepsilon$ to $\|\eta\|$). Due to its importance, this problem has recently gained increasing attention in the bandit community for regret minimization [20, 29, 18, 33, 39]. However it has not been addressed in the context of pure exploration. In this paper, we take a step towards bridging this gap by studying fixed-confidence Top-$m$ identification in the context of misspecified linear bandits. Our detailed contributions are as follows.

**Contributions.** (1) We derive a tractable lower bound on the sample complexity of any $\delta$-correct algorithm for the general Top-$m$ identification problem. (2) Leveraging this lower bound, we show that knowing an upper bound $\varepsilon$ to $\|\eta\|$ is necessary for adapting to the scale of misspecification, in the sense that any $\delta$-correct algorithm without such information cannot achieve a better sample complexity than that obtainable when no structure is available. (3) We design the first algorithm for Top-$m$ identification in misspecified linear bandits. We derive an upper bound to its sample complexity that holds for any $\delta \in (0, 1)$ and that matches our lower bound for $\delta \to 0$. Notably, our analysis reveals a nice adaptation to the value of $\varepsilon$, recovering state-of-the-art dependences in the linear case ($\varepsilon = 0$), where the sample complexity scales polynomially in $d$ and not in $K$, and in the unstructured case ($\varepsilon$ large), where only polynomial terms in $K$ appear. (4) We evaluate our algorithm on synthetic problems and real datasets from drug repurposing and recommendation system applications, while showing competitive performance with state-of-the-art methods.

**Related work.** While model misspecification has not been addressed in the pure exploration literature, several attempts to tackle this problem in the context of regret minimization exist. In [20], the authors show that, if $T$ is the learning horizon, for any bandit algorithm which enjoys $\mathcal{O}(d\sqrt{T})$ regret scaling on linear models, there exists a misspecified instance where the regret is necessarily linear. As a workaround, the authors design a statistical test based on sampling a subset of arms prior to learning to decide whether a linear or an unstructured bandit algorithm should be run on the data. Similar ideas are presented in [8], where the authors design a sequential test to switch online between linear and unstructured models. More recently, elimination-based algorithms [29, 39] and model selection methods [33, 18] have attracted increasing attention. Notably, these algorithms adapt to the amount of misspecification $\varepsilon$ *without* knowing it beforehand, at the cost of an additive linear term that scales with $\varepsilon$. Moreover, while best-arm identification has been the focus of many prior works in the realizable linear setting, some suggesting asymptotically-optimal algorithms [13, 21], Top-$m$ identification has been seldom studied in terms of problem-dependent lower bounds. Lower bounds for the unstructured Top-$m$ problem have been derived previously, focusing on explicit bounds [26], on getting the correct dependence in the problem parameters for any confidence $\delta$ [9, 37], or on asymptotic optimality (as $\delta \to 0$) [19]. Because of the combinatorial nature of the Top-$m$ identification problem, obtaining a tractable, tight, problem-dependent lower bound is not straightforward.

## 2 Setting

At successive stages $t \in \mathbb{N}$, the learner samples an arm $k_t \in [K]$ based on previous observations and internal randomization (a random variable $U_t \in [0,1]$) and observes a reward $X_t^{k_t}$. Let $\mathcal{F}_t := \sigma(\{U_1, k_1, X_1^{k_1}, \ldots, U_t, k_t, X_t^{k_t}, U_{t+1}\})$ be the $\sigma$-algebra associated with past sampled arms and rewards until time $t$. Then $k_t$ is a $\mathcal{F}_{t-1}$-measurable random variable. The reward $X_t^{k_t}$ is sampled from $\nu^{k_t}$ and is independent of all past observations, conditionally on $k_t$. We suppose that the noise is Gaussian with variance 1, such that the observation when pulling arm $k_t$ at time $t$ is $X_t^{k_t} \sim \mathcal{N}(\mu^{k_t}, 1)$. The mean vector $\mu = (\mu^k)_{k \in [K]} \in \mathbb{R}^K$ then fully describes the reward distributions.

In a misspecified linear bandit, each arm $k \in [K]$ is described by a feature vector $\phi_k \in \mathbb{R}^d$. The corresponding feature matrix is denoted by $A := [\phi_1, \phi_2, \ldots, \phi_K]^\top \in \mathbb{R}^{K \times d}$ and the maximum $\ell_2$-norm of these vectors is $L := \max_{k \in [K]} \|\phi_k\|_2$. We assume that the feature vectors span $\mathbb{R}^d$ (otherwise we could rewrite those vectors in a subspace of smaller dimension). We assume that the learner is provided with a set of realizable models

$$\mathcal{M} := \left\{ \mu \in \mathbb{R}^K \mid \exists \theta \in \mathbb{R}^d \; \exists \eta \in \mathbb{R}^K, \; \mu = A\theta + \eta \; \wedge \; \|\mu\|_\infty \le M \; \wedge \; \|\eta\|_\infty \le \varepsilon \right\}, \quad (1)$$

where $M, \varepsilon \in \mathbb{R}$ are known upper bounds on the $\ell^\infty$-norm of the mean[1] and misspecification vectors, respectively. Intuitively, $\mathcal{M}$ represents the set of bandit models whose mean vector $\mu$ is linear in the features $A$ only up to some misspecification $\eta$.

We consider Top-$m$ identification in the fixed-confidence setting. Given a confidence parameter $\delta \in (0,1)$, the learner is required to output the $m \in [K]$ arms of the unknown bandit model $\mu \in \mathcal{M}$ with highest means with probability at least $1 - \delta$. The strategy of a bandit algorithm designed for Top-$m$ identification can be decomposed into three rules: a *sampling* rule, which selects the arm $k_t$ to sample at a given learning round $t$ according to past observations; a *stopping* rule, which determines the end of the learning phase, and is a stopping time with respect to the filtration $(\mathcal{F}_t)_{t>0}$, denoted by $\tau_\delta$; finally, a *decision* rule, which returns a $\mathcal{F}_{\tau_\delta}$-measurable answer to the pure exploration problem. An answer is a set $\hat{S}_m \subseteq [K]$ with exactly $m$ arms: $|\hat{S}_m| = m$. In our context, the "$m$ best arms of $\mu$" might not be well defined since the set $S^\star(\mu) := \{k \in [K] \mid \mu^k \ge \max_{i \in [K]}^m \mu^i\}$[2] might contain more than $m$ elements if some arms have the same mean. Thus, let $\mathcal{S}_m(\mu) = \{S \subseteq S^\star(\mu) \mid |S| = m\}$ be the set containing all subsets of $m$ elements of $S^\star(\mu)$.

**Definition 1** ($\delta$-correctness). For $\delta \in (0,1)$, we say that an algorithm $\mathfrak{A}$ is $\delta$-correct on $\mathcal{M}$ if, for all $\mu \in \mathcal{M}$, $\tau_\delta < +\infty$ almost surely and $\mathbb{P}_\mu^\mathfrak{A}\left(\hat{S}_m \notin \mathcal{S}_m(\mu)\right) \le \delta$.

## 3 Tractable lower bound for the general Top-$m$ identification problem

Let $N_t^k$ denote the number of times arm $k$ has been sampled until time $t$ included. Suppose that the true model $\mu$ has exactly $m$ arms that are among the top-$m$, *i.e.*, that $|S^\star(\mu)| = m$ and $\mathcal{S}_m(\mu) = \{S^\star(\mu)\}$. Consider the following set of alternatives to $\mu$,

$$\Lambda_m(\mu) := \{\lambda \in \mathcal{M} \mid \mathcal{S}_m(\lambda) \cap \mathcal{S}_m(\mu) = \emptyset\},$$

that is, the set of all bandit models $\lambda$ in $\mathcal{M}$ where the top-$m$ arms of $\mu$ are not among the top-$m$ arms of $\lambda$. Note that, while we assumed that the set of top-$m$ arms in $\mu$ is unique, this might not be the case for $\lambda$. Define the event $E_{\tau_\delta} := \{\hat{S}_m \in \mathcal{S}_m(\mu)\}$ that the answer returned by the algorithm at $\tau_\delta$ is correct for $\mu$ and consider any $\delta$-correct algorithm $\mathfrak{A}$. Let us call KL the Kullback-Leibler divergence[3] and kl the binary relative entropy. Then, using the change-of-measure argument proposed in [19, Theorem 1], for any $\lambda \in \Lambda_m(\mu)$ and $\delta \le 1/2$,

$$\sum_{k \in [K]} \mathbb{E}_\mu^\mathfrak{A}[N_\tau^k] \, \mathrm{KL}\left(\mu^k, \lambda^k\right) \ge \mathrm{kl}\left(\mathbb{P}_\mu^\mathfrak{A}(E_{\tau_\delta}), \mathbb{P}_\lambda^\mathfrak{A}(E_{\tau_\delta})\right) \ge \mathrm{kl}(1 - \delta, \delta) \ge \log\left(\frac{1}{2.4\delta}\right),$$

---

[1]The restriction to $\|\mu\|_\infty \le M$ is required only for our analysis, while it can be safely dropped in practice.
[2]The expression $\max_{i \in S}^m f(i)$ denotes the $m^{th}$ maximal value in $\{f(i) \mid i \in S\}$.
[3]We abuse notation by denoting distributions in the same one-dimensional exponential family by their means.

where the second-last inequality follows from the $\delta$-correctness of the algorithm and the monotonicity of the function kl. This holds for any $\lambda \in \Lambda_m(\mu)$, so we have that

$$\mathbb{E}_{\mu}^{\mathfrak{A}}[\tau] \geq \left( \sup_{\omega \in \Delta_K} \inf_{\lambda \in \Lambda_m(\mu)} \sum_{k \in [K]} \omega^k \mathrm{KL}\left(\mu^k, \lambda^k\right) \right)^{-1} \log\left(\frac{1}{2.4\delta}\right), \tag{2}$$

with $\Delta_K := \{p \in [0,1]^K \mid \sum_{k=1}^K p_k = 1\}$ the simplex on $[K]$. We define the inverse complexity $H_\mu := \sup_{\omega \in \Delta_K} \inf_{\lambda \in \Lambda_m(\mu)} \sum_{k \in [K]} \omega^k \mathrm{KL}\left(\mu^k, \lambda^k\right)$. Computing that lower bound might be difficult: while the Kullback-Leibler is convex for Gaussians, the set $\Lambda_m(\mu)$ over which it is minimized is non-convex. Its description using $\mathcal{S}_m(\lambda)$ is combinatorial: we can write $\Lambda_m(\mu)$ as a union of convex sets, one for each subset of top-$m$ arms of $\lambda$, but this implies minimizing over $\binom{K}{m}$ sets, which is not practical. In order to rewrite this lower bound, we prove the following lemma in Appendix C.

**Lemma 1.** $\forall \mu, \lambda \in \mathbb{R}^K s.t. |S^\star(\mu)| = m, \mathcal{S}_m(\lambda) \cap \mathcal{S}_m(\mu) = \emptyset \Leftrightarrow \exists i \notin S^\star(\mu) \, \exists j \in S^\star(\mu), \lambda^i > \lambda^j$.

Lemma 1 allows us to go from an exponentially costly optimization problem, which implied minimizing over $\binom{K}{m}$ sets, to optimizing across $m(K - m)$ halfspaces. Therefore, by replacing the set of alternative models as derived in Lemma 1, the lower bound in Equation 2 can be rewritten in the following more convenient form :

**Theorem 1.** *For any $\delta \leq 1/2$, for any $\delta$-correct algorithm $\mathfrak{A}$ on $\mathcal{M}$, for any bandit instance $\mu \in \mathbb{R}^K$ such that $|S^\star(\mu)| = m$, the following lower bound holds on the stopping time $\tau_\delta$ of $\mathfrak{A}$ on instance $\mu$:*

$$\mathbb{E}_{\mu}^{\mathfrak{A}}[\tau_\delta] \geq \left( \sup_{\omega \in \Delta_K} \min_{i \notin S^\star(\mu)} \min_{j \in S^\star(\mu)} \inf_{\lambda \in \mathcal{M}: \lambda^i > \lambda^j} \sum_{k \in [K]} \omega^k \mathrm{KL}\left(\mu^k, \lambda^k\right) \right)^{-1} \log\left(\frac{1}{2.4\delta}\right).$$

Computing the lower bound now requires performing one maximization over the simplex (which can be still hard), and $m(K - m)$ minimizations over half-spaces $\{\lambda \in \mathcal{M} : \lambda^i > \lambda^j\}$, where $(i, j) \in (\mathcal{S}^\star(\mu))^c \times \mathcal{S}^\star(\mu)$. The minimizations are convex optimization problems and can be solved efficiently. Our algorithm inspired from that bound will need to perform only those minimizations.

Note that a lower bound for Top-$m$ identification using the cited change-of-measure argument has been obtained in [26]. Aiming to be more explicit, it relies on alternative models where one of the best arms is switched with the $(m + 1)^{th}$ best one (or one of the $K - m$ worst ones with the $m^{th}$ best one). These models are a strict subset of $\Lambda_m(\mu)$. Hence this bound is not as tight as the one in Theorem 1, which is why the algorithm we detail in the next sections will rely on the latter instead.

Note that with $\varepsilon = 0$ and $m = 1$, this lower bound is exactly the one for best arm identification in perfectly linear models [17]. As the misspecification $\varepsilon$ grows, the set $\mathcal{M}$ becomes larger and so does the set of alternative models $\Lambda_m(\mu)$, thus the lower bound grows. In the limit $\varepsilon \to +\infty$, the model becomes the same as the unstructured model. We show that in fact the lower bound becomes exactly equal to the unstructured lower bound as soon as $\varepsilon > \varepsilon_\mu$, a finite value.

**Lemma 2.** *There exists $\varepsilon_\mu \in \mathbb{R}$ with $\varepsilon_\mu \leq \max_k \mu^k - \min_k \mu^k$ such that if $\varepsilon > \varepsilon_\mu$, then the lower bound of Theorem 1 is equal to the unstructured top-$m$ lower bound.*

The proof is in Appendix C. It considers finitely supported distributions over $\Lambda_m(\mu)$ that realize the equilibrium in the max-min game of the lower bound. As soon as one of these equilibrium distributions for the unstructured problem has its whole support in the misspecified model, the two complexities are equal.

## 3.1 Adaptation to unknown misspecification is impossible

We now make an important observation: knowing that a problem is misspecified without knowing an upper bound $\varepsilon$ on $\|\eta\|_\infty$ is the same as not knowing anything about the structure of that problem.

The lower bound of Equation (2) is a function of the set $\mathcal{M}$ of realizable models $\mu$. Let $B(\mu, \delta, \mathcal{M})$ be the right-hand side of that equation, such that $\mathbb{E}_{\mu}^{\mathfrak{A}}[\tau_\delta] \geq B(\mu, \delta, \mathcal{M})$ for any algorithm $\mathfrak{A}$ which is $\delta$-correct on $\mathcal{M}$. Suppose that we have $\mathcal{M}_1 \subseteq \mathcal{M}$, a subset of the model, for which we would like to have lower sample complexity (possibly at the cost of a higher sample complexity on $\mathcal{M} \setminus \mathcal{M}_1$). If

---

**Algorithm 1** MISLID

---
**Require:** Set of models $\mathcal{M}$, online learner $\mathcal{L}$, stopping thresholds $\{\beta_{t,\delta}\}_{t\geq1}$

    Compute a sequence of arms $k_1,\ldots,k_{t_0}$ such that $\sum_{t=1}^{t_0}\phi_{k_t}\phi_{k_t}^T \succeq 2L^2 I_d$      // INITIALIZATION
    **for** $t=1,\ldots,t_0$ **do**
        Pull $k_t$, receive $X_t^{k_t}$, and set $\omega_t \leftarrow e_{k_t}$      // PULL SPANNER
    **end for**
    Compute empirical mean $\widehat{\mu}_{t_0}$ and its projection $\tilde{\mu}_{t_0} \leftarrow \arg\min_{\lambda\in\mathcal{M}} \|\lambda - \widehat{\mu}_{t_0}\|_{D_{N_{t_0}}}^2$
    **for** $t=t_0+1, t_0+2,\ldots,$ **do**
        **if** $\inf_{\lambda\in\Lambda_m(\tilde{\mu}_{t-1})}\|\tilde{\mu}_{t-1}-\lambda\|_{D_{N_{t-1}}}^2 > 2\beta_{t-1,\delta}$ **then**      // STOPPING RULE
            Stop and return $\mathcal{S}_m^\star(\tilde{\mu}_{t-1})$
        **end if**
        Obtain $\omega_t$ from $\mathcal{L}$
        Compute closest alternative: $\lambda_t \leftarrow \arg\min_{\lambda\in\Lambda_m(\tilde{\mu}_{t-1})}\|\tilde{\mu}_{t-1}-\lambda\|_{D_{\omega_t}}^2$
        Update $\mathcal{L}$ with gain $g_t : \omega \mapsto \sum_{k\in[K]}\omega^k\left(|\tilde{\mu}_{t-1}^k - \lambda_t^k| + \sqrt{c_{t-1}^k}\right)^2$      // UPDATE LEARNER
        Pull $k_t \sim \omega_t$ and receive reward $X_t^{k_t}$      // ACTION SAMPLING
        Update $\widehat{\mu}_t$ and compute projection $\tilde{\mu}_t \leftarrow \arg\min_{\lambda\in\mathcal{M}}\|\lambda-\widehat{\mu}_t\|_{D_{N_t}}^2$      // ESTIMATION
    **end for**

---

$\mathcal{M}$ is the misspecified linear model with deviation $\varepsilon$, let us say that $\mathcal{M}_1$ is the set of problems with deviation lower than $\varepsilon_1 < \varepsilon$; that is, we want the algorithm to be faster on more linear models. This is not achievable. The lower bound states that it is *not* possible for an algorithm to have lower sample complexity on $\mathcal{M}_1$ while being $\delta$-correct on $\mathcal{M}$. On every $\mu\in\mathcal{M}$, the lower bound is $B(\mu,\delta,\mathcal{M})$.

An algorithm cannot adapt to the deviation to linearity: it has to use a parameter $\varepsilon$ set in advance, and its sample complexity will depend on that $\varepsilon$, not on the actual deviation of the problem. Note that this observation does not contradict recent results for regret minimization [e.g., 29, 39], which show that adapting to an unknown scale of misspecification is possible. In fact, such results involve a "weak" form of adaptivity, where the algorithms provably leverage the linear structure at the price of suffering an additive *linear* regret term of order $\mathcal{O}(\varepsilon\sqrt{d}T)$, where $T$ is the learning horizon. Since the counterpart of $\delta$-correctness for regret minimization is "the algorithm suffers sub-linear regret in $T$ for all instances of the given family", this implies that algorithms with such "weak" adaptivity loose this important property of consistency.

## 4 The MISLID algorithm

We introduce MISLID (Misspecified Linear Identification), an algorithm to tackle misspecification in linear bandit models for fixed-confidence Top-$m$ identification. We describe the algorithm in Section 4.1, while in Section 4.2 we report its sample complexity analysis.

### 4.1 Algorithm

The pseudocode of MISLID is outlined in Algorithm 1. On the one hand, the design of MISLID builds on top of recent approaches for constructing pure exploration algorithms from lower bounds [12, 13, 43, 21]. On the other hand, its main components and their analysis introduce several technical novelties to deal with misspecified Top-$m$ identification, that might be of independent interest for other settings. We describe these components below. Let us define $D_v := \mathrm{diag}(v^1, v^2, \ldots, v^K)$ for any vector $v\in\mathbb{R}^K$, and $V_t := \sum_{s=1}^t \phi_{k_s}\phi_{k_s}^\top$.

**Initialization phase.** MISLID starts by pulling a deterministic sequence of $t_0$ arms that make the minimum eigenvalue of the resulting design matrix $V_{t_0}$ larger than $2L^2$. Since the rows of $A$ span $\mathbb{R}^d$, such sequence can be easily found by taking any subset of $d$ arms that span the whole space (e.g., by computing a barycentric spanner [4]) and pulling them in a round robin fashion until the desired condition is met. This is required to make the design matrix invertible. While the literature typically avoid this step by regularizing (*e.g.*, [1]), in our misspecified setting it is crucial not to do

so to obtain tight concentration results for the estimator of $\mu$, as explained in the next paragraph. See Appendix D.1 for a discussion of the length $t_0$ of that initialization phase.

**Estimation.** At each time step $t \geq t_0$, MISLID maintains an estimator $\tilde{\mu}_t$ of the true bandit model $\mu$. This is obtained by first computing the empirical mean $\widehat{\mu}_t$, such that $\widehat{\mu}_t^k = \frac{1}{N_t^k} \sum_{s=1}^t \mathbb{1}\{k_s = k\} X_s^{k_s}$, and then projecting it onto the family of realizable models $\mathcal{M}$ according to the $D_{N_t}$-weighted norm, i.e., $\tilde{\mu}_t := \arg\min_{\lambda \in \mathcal{M}} \|\lambda - \widehat{\mu}_t\|_{D_{N_t}}^2$. Since each $\lambda \in \mathcal{M}$ can be decomposed into $\lambda = A\theta' + \eta'$ for some $\theta' \in \mathbb{R}^d$ and $\eta' \in \mathbb{R}^K$, this can be solved efficiently as the minimization of a quadratic objective in $K + d$ variables subject to the linear constraints $\|\eta'\|_\infty \leq \varepsilon$ and $\|A\theta' + \eta'\|_\infty \leq M$. The second constraint is only required for the analysis, while it often has a negligible impact in practice. Thus, we shall drop it in our implementation, which yields two independent optimization problems for the projection $\tilde{\mu}_t = A\tilde{\theta}_t + \tilde{\eta}_t$: one for $\tilde{\theta}_t$, whose solution is available in closed form as the standard least-squares estimator $\tilde{\theta}_t = \hat{\theta}_t := V_t^{-1} \sum_{s=1}^t X_s^{k_s} \phi_{k_s}$, and one for $\tilde{\eta}_t$, which is another quadratic program with $K$ variables (see Appendix D).

A crucial component in the concentration of these estimators, and a key novelty of our work, is the adoption of an orthogonal parametrization of mean vectors. In particular, we leverage the following observation: any mean vector $\mu = A\theta + \eta$ can be equivalently represented, at any time $t$, as $\mu = A\theta_t + \eta_t$, where $\theta_t = V_t^{-1} \sum_{s=1}^t \mu^{k_s} \phi_{k_s}$ is the orthogonal projection (according to the design matrix $V_t$) of $\mu$ onto the feature space and $\eta_t = \mu - A\theta_t$ is the residual. Then, it is possible to show that $\|\hat{\theta}_t - \theta_t\|_{V_t}^2$ is *exactly* the self-normalized martingale considered in [1] and, thus, it enjoys the *same* bound we have in linear bandits with no misspecification (refer to Appendix B). This is an important advantage over prior works [29, 44] that, in order to concentrate $\hat{\theta}_t$ to $\theta$, need to inflate the concentration rate by a factor $\varepsilon^2 t$, which often makes the bound too large to be practical for misspecified models with $\varepsilon \gg 0$. It allows us to also avoid superlinear terms of the form $\epsilon^2 t \log(t)$ which are present in related works and which would prevent us from deriving good problem-dependent guarantees.

**Stopping rule.** MISLID uses the standard stopping rule adopted in most existing algorithms for pure exploration [19, 12, 36]. What makes it peculiar is the definition of the thresholds $\beta_{t,\delta}$. MISLID requires a careful combination of concentration inequalities for (1) linear bandits, to make the algorithm adapt well to linear models with low $\varepsilon$, and (2) unstructured bandits, to guarantee asymptotic optimality. The precise definition of $\beta_{t,\delta}$ is shown in the following result.

**Lemma 3** (MISLID is $\delta$-correct). *Let $W_{-1}$ be the negative branch of the Lambert W function and let $\overline{W}(x) = -W_{-1}(-e^{-x}) \approx x + \log x$. For $\delta \in (0,1)$, define*

$$\beta_{t,\delta}^{\text{uns}} := 2K\overline{W}\left(\frac{1}{2K}\log\frac{2e}{\delta} + \frac{1}{2}\log(8eK\log t)\right), \tag{3}$$

$$\beta_{t,\delta}^{\text{lin}} := \frac{1}{2}\left(4\sqrt{t}\varepsilon + \sqrt{2}\sqrt{1 + \log\frac{1}{\delta} + \left(1 + \frac{1}{\log(1/\delta)}\right)\frac{d}{2}\log\left(1 + \frac{t}{2d}\log\frac{1}{\delta}\right)}\right)^2. \tag{4}$$

*Then, for the choice $\beta_{t,\delta} := \min\{\beta_{t,\delta}^{\text{uns}}, \beta_{t,\delta}^{\text{lin}}\}$, MISLID is $\delta$-correct.*

This result is a simple consequence of two (linear and unstructured) concentration inequalities. See Appendix F.

**Sampling strategy and online learners.** The sampling strategy of MISLID aims at achieving the optimal sample complexity from the lower bound in Theorem 1. As popularized by recent works [12, 13, 43], instead of relying on inefficient max-min oracles to repeatedly solve the optimization problem of Theorem 1 [17, 21], we compute it incrementally by employing no-regret online learners. At each step $t$, the learner $\mathcal{L}$ plays a distribution over arms $\omega_t \in \Delta_K$ and it is updated with a gain function $g_t$ whose precise definition will be specified shortly. Then, MISLID directly samples the next arm to pull from the distribution $\omega_t$, instead of using tracking as in the majority of previously mentioned works. Similarly to what was recently shown by [40] for regret minimization in linear bandits, sampling will be crucial in our analysis to reduce dependencies on $K$ and, in particular, to obtain only logarithmic dependencies in the realizable linear case.

Regarding the choice of $\mathcal{L}$, two important properties are worth mentioning. First, MISLID requires only a *single* learner, while existing asymptotically optimal algorithms for pure exploration [12, 13]

need to allocate one learner for each possible answer. Since the number of answers is $\binom{K}{m}$, a direct extension of these algorithms to the Top-$m$ setting would yield an impractical method with exponential (in $K$) number of learners, hence space complexity, and possibly sample complexity.[4] Second, the choice of $\mathcal{L}$ is highly flexible since any learner that satisfies the following property suffices.

**Definition 2** (No-regret learner). *A learner $\mathcal{L}$ over $\Delta_K$ is said to be no-regret if, for any $t \geq 1$ and any sequence of gains $\{g_s(\omega)\}_{s \leq t}$ bounded in absolute value by $B \in \mathbb{R}^+$, there exists a positive constant $C_{\mathcal{L}}(K, B)$ such that $\max_{w \in \Delta_K} \sum_{s=1}^{t} \left(g_s(w) - g_s(w_s)\right) \leq C_{\mathcal{L}}(K, B)\sqrt{t}$.*

Examples of algorithms in this class are Exponential Weights [7] and AdaHedge [15]. The latter shall be our choice for the implementation since it does not use $B$ as a parameter but adapts to it, and thus does not suffer from a possibly loose bound on $B$.

**Optimistic gains.** Finally, we need to specify how the gains $g_t$ are computed. Clearly, if $\mu$ were known, one would directly use $g_t : \omega \mapsto \inf_{\lambda \in \Lambda_m(\mu)} \|\mu - \lambda\|^2_{D_\omega}$. Since $\mu$ is unknown and must be estimated, we set $g_t(\omega)$ to an *optimistic* proxy for that quantity. In particular, we choose a sequence of bonuses $\{c_t^k\}_{t \geq t_0, k \in [K]}$ such that, with high probability, $g_t(\omega_t) := \sum_{k \in [K]} \omega_t^k \left(|\tilde{\mu}_{t-1}^k - \lambda_t^k| + \sqrt{c_{t-1}^k}\right)^2 \geq \inf_{\Lambda \in \lambda_m(\mu)} \|\mu - \lambda\|^2_{D_{\omega_t}}$, for $\lambda_t := \arg\min_{\lambda \in \Lambda_m(\tilde{\mu}_{t-1})} \|\tilde{\mu}_{t-1} - \lambda\|^2_{D_{\omega_t}}$. As for the stopping thresholds, we construct $c_t^k$ by a careful combination of structured and unstructured concentration bounds:

$$c_t^k := \min\left\{ 8(LK+1)^2\varepsilon^2 + 4\alpha_{t^2}^{\text{lin}}\|\phi_k\|^2_{V_t^{-1}}, \frac{2\alpha_{t^2}^{\text{uns}}}{N_t^k}, 4M^2 \right\},$$

where $\alpha_t^{\text{uns}} := \beta_{t,1/(5t^3)}^{\text{uns}}$ and $\alpha_t^{\text{lin}} := \log(5t^2) + d\log(1 + t/(2d))$. We show in Appendix F that this choice of $c_t^k$ suffices to guarantee optimism with high probability.

## 4.2 Sample complexity

**Theorem 2.** MISLID *has expected sample complexity $\mathbb{E}_\mu[\tau_\delta] \leq T_0(\delta) + 2$, where $T_0(\delta)$ is the solution to the equation in $t$*

$$\beta_{t,\delta} \geq tH_\mu + \widehat{\mathcal{O}}\left( \min\{tK^2\varepsilon^2 + d\sqrt{t}\ell_t, \sqrt{K}t\ell_t\}; \log K\sqrt{t}; \sqrt{\min\{tK^2\varepsilon^2 + d\ell_t, K\ell_t\}\log(1/\delta)} \right), \tag{5}$$

*where $\ell_t := \log t$, $H_\mu$ is the inverse complexity appearing in the lower bound (see Equation 2), and $\widehat{\mathcal{O}}(a; b; c)$ represent a sum of terms, each of which is $\mathcal{O}$ of one of the expressions shown.*

See Appendix F for the proof. Since $\beta_{t,\delta}^{\text{uns}} \approx \log(1/\delta)$ for small $\delta$, $T_0(\delta) = H_\mu^{-1}\log(1/\delta) + C_\mu o(\log(1/\delta))$, where $C_\mu$ is a problem-dependent constant. Then $\liminf_{\delta \to 0} \mathbb{E}_\mu[\tau_\delta]/\log(1/\delta) = \liminf_{\delta \to 0} T_0(\delta)/\log(1/\delta) = H_\mu^{-1}$ and thus the upper bound matches the lower bound in that limit: MISLID is asymptotically optimal. The only polynomial factors in $K$ are in a minimum with a term that depends on $\varepsilon$. In the linear setting, when $\varepsilon = 0$, we have only logarithmic (and no polynomial) dependence on the number of arms, which is on par with the state of the art [40, 21, 27]. Moreover, the bound exhibits an adaptation to the value of $\varepsilon$. If $\varepsilon$ is small, then the minimums in $\beta_{t,\delta}$ and in the inequality (5) are equal to the "linear" values which involve $K\varepsilon$ and $d$ instead of $K$. As $\varepsilon$ grows, the upper bound transitions to terms matching the optimal unstructured bound.

**Decoupling the stopping and sampling analyses.** Our analysis decomposes into two parts: first, a result on the stopping rule, then, a discussion of the sampling rule. The algorithm is shown to verify that, under a favorable event, if it does not stop at time $t$,

$$2\beta_{t,\delta} \geq \inf_{\lambda \in \Lambda_m(\mu)} \|\mu - \lambda\|^2_{D_{N_t}} - \mathcal{O}(\sqrt{t}) \geq 2tH_\mu - \mathcal{O}(\sqrt{t}).$$

The sample complexity result is a consequence of that bound on $t$. The first inequality is due solely to the stopping rule, and the second one only to the sampling mechanism. The expression

---

[4]The fact that the optimization problem of the lower bound decomposes into $m(K - m)$ minimizations does not reduce the number of possible answers, which is still combinatorial in $K$.

$\inf_{\lambda \in \Lambda_m(\mu)} \|\mu - \lambda\|^2_{D_{N_t}}$ does not feature any variable specific to the algorithm: we can combine any stopping rule and any sampling rule, as long as they each verify the corresponding inequality.

**A more aggressive optimism.** The optimistic gains that we have chosen, $g_t(\omega) = \sum_{k \in [K]} \omega^k (|\tilde{\mu}^k_{t-1} - \lambda^k_t| + \sqrt{c^k_{t-1}})^2$, are tuned to ensure asymptotic optimality (with a factor 1 in the leading term). If we instead accept to be asymptotically optimal up to a factor 2, we can use the gains $g_t(\omega) = \sum_{k \in [K]} \omega^k \left((\tilde{\mu}^k_{t-1} - \lambda^k_t)^2 + c^k_{t-1}\right)$. When using those, the learner takes decisions which are much closer to those it would take if using the empirical gains $\sum_{k \in [K]} \omega^k (\tilde{\mu}^k_{t-1} - \lambda^k_t)^2$ and the theoretical bound, while worse in the leading factor, has better lower order terms. The aggressive optimism sometimes has significantly better practical performance (see Experiment (C) in Figure 1).

## 5 Experimental evaluation

Since our algorithm is the first to apply to Top-$m$ identification in misspecified linear models, we compare it against an efficient linear algorithm, LinGapE [42] (that is, its extension to Top-$m$ as described in [35], which coincides with LinGapE for $m = 1$), and an unstructured one, LUCB [23]. In all experiments, we consider $\delta = 5\%$. [5] For each algorithm, we show boxplots reporting the average sample complexity on the $y$-axis, and the error frequency $\hat{\delta}$ across 500 (resp. 100) repetitions for simulated (resp. real-life) instances rounding up to the $5^{th}$ decimal place. Individual outcomes are shown as gray dots. It has frequently been noted in the fixed-confidence literature that stopping thresholds which guarantee $\delta$-correctness tend to be too conservative and to yield empirical error frequencies that are actually much lower than $\delta$. Moreover, these thresholds are different from linear to unstructured models. In order to ensure a good trade-off between performance and computing speed, and fairness between tested algorithms, we use a heuristic value for the stopping rule $\beta_{t,\delta} := \ln((1 + \ln(t + 1))/\delta)$ unless otherwise specified. For each experiment, we report the number of arms ($K$), the dimension of features ($d$), the size of the answer ($m$), the misspecification ($\varepsilon$) and the gap between the $m^{th}$ and $(m + 1)^{th}$ best arms ($\Delta := \max^m_{a \in [K]} \mu^a - \max^{m+1}_{b \in [K]} \mu^b$). The computational resources used, data licenses and further experimental details can be found in Appendix G.

**(A) Simulated misspecified instances.** ($K = 10$, $d = 5$, $m = 3$, $\varepsilon \in \{0, 5\}$, $\Delta \approx 0.28$) First, we fix a linear instance $\mu := A\theta$ by randomly sampling the values of $\theta \in \mathbb{R}^d$ and $A \in \mathbb{R}^{K \times d}$ from a zero-mean Gaussian distribution, and renormalizing them by their respective $\ell_\infty$ norm. Then, for $\varepsilon \in \{0, 5\}$, we build a misspecified linear instance $\mu_\varepsilon = A\theta + \eta_\varepsilon$, such that, if (4) is the index of the fourth best arm, $\forall k \neq (4), \eta^k_\varepsilon = 0$, and $\eta^{(4)}_\varepsilon = \varepsilon$. Note that any value of $\varepsilon < \Delta$ does not switch the third and fourth arms in the set of best arms of $\mu_\varepsilon$, contrary to values greater than $\Delta$. The greater $\varepsilon$ is, the more different the answers from the linear and misspecified models are. This experiment was inspired by [20], where a similar model is used to prove a lower bound in the setting of regret minimization. See the leftmost two plots on Figure 1. As expected, LUCB is always $\delta$-correct, but suffers from a significantly larger sample complexity than its structured counterparts. Moreover, LinGapE does not preserve the $\delta$-correctness under large misspecification level $\varepsilon = 5$ (with error rate $\hat{\delta} = 0.96$), which illustrates the effect of $\varepsilon$ on the answer set. Note that it is not due to the choice of stopping threshold, as running it with the theoretically-supported threshold derived in [1] also yields an empirical error rate $\hat{\delta} = 1$. MISLID proves to be competitive against LinGapE. Note that the case $\varepsilon = 0$ is a perfectly linear instance. See Table 2 in Appendix for numerical results for algorithms LinGapE and MISLID.

**(B) Discrepancy between user-selected $\varepsilon$ and true $\|\eta\|$.** ($K = 15$, $d = 8$, $m = 3$, $\varepsilon \in \{0.5, 1, 2\}$, $\Delta \approx 0.4$) MISLID crucially relies on a user-provided upper bound on the scale of deviation from linearity. We test its robustness against perturbations to the input value $\varepsilon$ compared to the value $\varepsilon^\star := \|\eta\|_\infty$ in the misspecified model $\mu := A\theta + \eta$. Values are sampled randomly for $\theta, A, \eta$, and the associated vectors are normalized by their $\ell^\infty$ norm (for $\eta$, by $\|\eta\|_\infty / \varepsilon^\star$), where $\varepsilon^\star = 1 > \Delta$ is the true deviation to linearity. The results, shown in the third plot of Figure 1, display the behavior predicted by Lemma 2. Indeed, as the user-provided value $\varepsilon$ increases, the associated sample complexity increases as well. The plateau in sample complexity when $\varepsilon$ is large enough is noticeable. Cases $\varepsilon \in \{1, 2\}$ display a sample complexity close to that of unstructured bandits.

---

[5] All the code and scripts are available at `https://github.com/clreda/misspecified-top-m`.

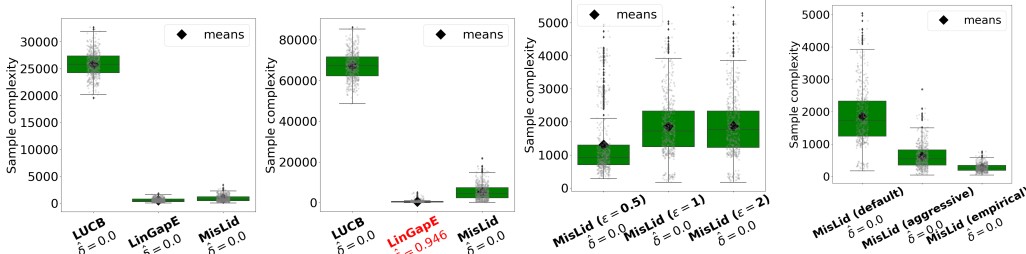

Figure 1: Experiment (A) for $\varepsilon \in \{0, 5\}$ (*first two from the left*). Experiment (B) with $\varepsilon \in \{0.5, 1, 2\}$. Experiment (C) to compare different optimistic gains (*right*).

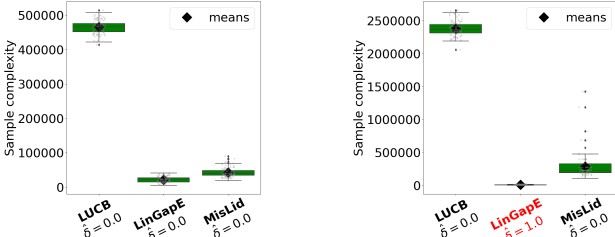

Figure 2: Experiment (D) for drug repurposing in epilepsy (*left*). Experiment (E) for online recommendation.

**(C) Comparing different optimisms.** ($K = 15$, $d = 8$, $m = 3$, $\varepsilon = 1$, $\Delta \approx 0.4$) We use the same bandit model as in Experiment (B), and use $\varepsilon = \varepsilon^\star = 1$. We compare the aggressive optimism described in Section 4.2, no optimism (that is, $\forall k \in [K], \forall t > 0, c_t^k = 0$), and the default optimistic gains given in Section 4.1. See the rightmost plot in Figure 1. The algorithm with no optimism is denoted "empirical", and is significantly faster than the optimistic variants.

**(D) Application to drug repurposing.** ($K = 10$, $d = 5$, $m = 5$, $\hat\varepsilon \approx 0.02$, $\Delta \approx 0.062$) We use the drug repurposing problem for epilepsy proposed by [35] to investigate the practicality of our method. In order to speed up LUCB, we consider the PAC version of Top-$m$ identification, choosing as stopping threshold $0.06 \approx \Delta$, such that the algorithm stops earlier while returning the exact set of $m$ best arms. Following [34, Appendix F.4], we extract a linear model from the data by fitting a neural network and taking the features learned in the last layer. We compute $\varepsilon$ as the $\ell^\infty$ norm of the difference between the predictions of this linear model and the average rewards from the data, which yields $\hat\varepsilon = 0.02$. Since the misspecification is way below the minimum gap, and the linear model thus accurately fits the data, the results (leftmost plot in Figure 2) show that MisLid and LinGapE perform comparably on this instance. Moreover, both are an order of magnitude better than an unstructured bandit algorithm sample complexity-wise. Please refer to Table 3 in Appendix for numerical results for LinGapE and MISLID.

**(E) Application to a larger instance of online recommendation.** ($K = 103$, $d = 8$, $m = 4$, $\hat\varepsilon \approx 0.206$, $\Delta \approx 0.022$) As in Experiment (D), a linear representation is extracted for an instance of online recommendation of music artists to users (Last.fm dataset [6]). We compute a proxy for $\varepsilon$ and feed the value $\Delta$ to the stopping threshold in LUCB in a similar fashion. Differently from Experiment (D), this yields a misspecification that is much larger than the minimum gap. To improve performance on these instances, we modified MISLID. To reduce the sample complexity, we use empirical gains instead of optimism. To reduce the computational complexity, we check the stopping rule infrequently (on a geometric grid) and use only a random subset of arms in each round to compute the sampling rule (see Appendix G for details and an empirical comparison to the theoretically supported MISLID). See the rightmost plot in Figure 2. This plot particularly illustrates our introductory claim: an unstructured bandit algorithm is $\delta$-correct, but too slow in practice for misspecified instances, whereas the guarantee on correctness for a linear bandit does not hold anymore on these models with large misspecification. Numerical results for LinGapE and MISLID are listed in Table 3 in Appendix.

# 6 Discussion

We have designed the first algorithm to tackle misspecification in fixed-confidence Top-$m$ identification, which has applications in online recommendation. However, the algorithm relies exclusively on the features provided in the input data, and as such might be subjected to bias and lack of fairness in its recommendation, depending on the dataset. The proposed algorithm can be applied to misspecified models which deviate from linearity (*i.e.*, $\varepsilon > 0$), encompassing unstructured settings (for large values of $\varepsilon$) and linear models (*i.e.*, $\varepsilon = 0$).

Our tests on variants of our algorithm suggest that the optimistic estimates have a big influence on the sample complexity. Removing the optimism completely and using the empirical gains leads to the best performance. We conjecture that other components of the algorithm like the learner are conservative enough for the optimism to be superfluous. The main limitation of our method is its computational complexity: at each round, $\mathcal{O}(Km)$ convex optimization problems need to be solved for both the sampling and stopping rules, which can be expensive if the number of arms is large. However, the "interesting" arms are much less numerous and we observed empirically that the sample complexity is not increased significantly if we consider only a few arms. In general, theoretically supported methods to replace the alternative set by computationally simpler approximations would greatly help in reducing the computational cost of our algorithm.

Since the sampling of our algorithm is designed to minimize a lower bound, we can expect it to suffer from the same shortcomings as that bound. It is known that the bound in question does not capture some lower order (in $1/\delta$) effects, in particular those due to the multiple-hypothesis nature of the test we perform, which can be very large for small times. Work to take these effects into account to design algorithms has started recently [24, 25, 41] and we believe that it is an essential avenue for further improvements in misspecified linear identification.

## Acknowledgments and Disclosure of Funding

Clémence Réda was supported by the "Digital health challenge" Inserm-CNRS joint program, the French Ministry of Higher Education and Research [ENS.X19RDTME-SACLAY19-22], and the French National Research Agency [ANR-19-CE23-0026-04] (BOLD project).

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
