# Appendix

## Table of Contents

# A   Notation

Table 1: Notation table

| Name | Description |
| --- | --- |
| $d \in \mathbb{N}^*$ | Dimension of the feature vectors |
| $K \in \mathbb{N}^*$ | Number of arms |
| $[K] := \{1, 2, \ldots, K\}$ | Enumeration |
| $m \in [K-1]$ | Number of best arms to return |
| $\mathbb{1}\{c\}$ | Kronecker's symbol, equal to 1 iff. claim $c$ is true |
| $\varepsilon \in \mathbb{R}^{*+}$ | Upper bound on the $\ell_\infty$ norm of the deviation to linearity |
| $M \in \mathbb{R}^{*+}$ | Upper bound on the $\ell_\infty$ norm on the mean vector |
| $L \in \mathbb{R}^{*+}$ | Upper bound on the $\ell_2$ norm on the arm feature vectors |
| $\delta \in (0,1)$ | Upper bound for the probability of error in identification |
| $e_k \in \mathbb{R}^k, k \in \mathbb{N}$ | $k^{th}$ vector of the canonical basis of $\mathbb{R}^k$ |
| $\Delta_K = \{p \in [0,1]^K \mid \sum_{k=1}^K p^k = 1\}$ | Set of probability distributions over finite set of size $K$ |
| $\phi_k \in \mathbb{R}^d, k \in [K]$ | Feature vector for arm $k$ |
| $A = [\phi_1, \phi_2, \ldots, \phi_K]^\top \in \mathbb{R}^{K \times d}$ | Feature matrix of arm contexts |
| $\Delta_K = \{p \in [0,1]^K \mid \sum_{k=1}^K p^k = 1\}$ | Set of probability distributions on finite set of size $K$ |
| $V_\omega := \sum_{k \le K} \omega_k \phi_k \phi_k^\top, \omega \in \Delta_K$ | Design matrix associated with $\omega$ |
| $V_t := \sum_{s \le t} \phi_{k_s} \phi_{k_s}^\top, t > 0$ | Design matrix at time $t$ |
| $\mathcal{M} \subset \mathbb{R}^K$ | Set of realizable models: $\{\mu \in \mathbb{R}^K \mid \exists \theta \in \mathbb{R}^d, \eta \in \mathbb{R}^K : \mu = A\theta + \eta, \|\mu\|_\infty \le M, \|\eta\|_\infty \le \varepsilon\}$ |
| $\mu \in \mathcal{M}$ | True mean vector: $\mu = A\theta + \eta$ |
| $N_t^k \in \mathbb{N}, k \in [K], t > 0$ | Number of times arm $k$ has been sampled until time $t$ included |
| $N_t = [N_t^1, N_t^2, \ldots, N_t^K]^\top \in \mathbb{N}^K$ | Vector of numbers of samplings for each arm at time $t$ included |
| $D_N \in \mathbb{R}^{K \times K}, N \in \mathbb{R}^K$ | Diagonal matrix with coefficients $N^1, N^2, \ldots, N^K$ |
| $k_s, s > 0$ | Arm sampled at time $s$ |
| $X_s^k, s > 0, k \in [K]$ | Reward observed at time $s$ from arm $k$ |
| $\tau_\delta, \delta \in (0,1)$ | Stopping time under $\delta$-correctness |
| $E_{\tau_\delta}$ | Event on $\delta$-correctness: $E_{\tau_\delta} := \left\{ \hat{S}_m \in \mathcal{S}_m(\mu) \right\}$ |
| $\widehat{\mu}_t \in \mathbb{R}^K, t > 0$ | Empirical mean vector at time $t$: $\widehat{\mu}_t^a := \frac{1}{N_t^a} \sum_{s \le t} X_s^a \mathbb{1}\{k_s = a\}$ |
| $\tilde{\mu}_t \in \mathbb{R}^K, t > 0$ | Projection of $\widehat{\mu}_t$ onto set $\mathcal{M}$ at time $t$ |
| $\hat{S}_m \subseteq [K], m \in [K-1]$ | Answer to Top-$m$ identification as returned by the algorithm |
| $S^\star(\mu) \subseteq [K], \mu \in \mathbb{R}^K$ | Set of best arms compared to the $m^{th}$ greatest mean: $S^\star(\mu) := \left\{ k \in [K] \mid \mu^k \ge \max_{i \in [K]}^m \mu^i \right\}$ |
| $\mathcal{S}_m(\mu), \mu \in \mathcal{M}, m \in [K-1]$ | Set of all subsets of size $m$ in $S^\star(\mu)$: $\mathcal{S}_m(\mu) := \{S \subseteq S^\star(\mu) \mid |S| = m\}$ |
| $\Lambda_m(\mu), \mu \in \mathcal{M}$ | Set of alternative models to model $\mu$: $\Lambda_m(\mu) := \{\lambda \in \mathcal{M} \mid \mathcal{S}_m(\lambda) \cap \mathcal{S}_m(\mu) = \emptyset\}$ |
| $H_\mu, \mu \in \mathcal{M}$ | Inverse complexity constant: $H_\mu := \sup_{\omega \in \Delta_K} \inf_{\lambda \in \Lambda_m(\mu)} \sum_{k \in [K]} \omega^k \mathrm{KL}(\mu^k, \lambda^k)$ |
| KL | Kullback-Leibler divergence |
| kl | Binary relative enthropy |
| $W_{-1}$ | Negative branch of the Lambert $W$ function |
| $\overline{W} : x \mapsto -W_{-1}(-e^{-x})$ | |
| $\mathcal{L}$ | Learner algorithm |
| $g_t(\omega), \omega \in \mathbb{R}^K, t > 0$ | Gains fed to the learner at time $t$ |
| $c_t^k, k \in [K], t > 0$ | Optimistic bonus, such that $(\tilde{\mu}_t^k - \mu^k)^2 \le c_t^k$ for any $k \in [K]$ and large enough $t > 0$, with high probability |

Please refer to Table 1. Moreover, if $\omega \in \mathbb{R}^K$, at $t > 0$, we also introduce the following notation related to orthogonal parameterizations (see Appendix B):

- $A_\omega := D_\omega^{1/2} A \in \mathbb{R}^{K \times d}$.
- $P_\omega := A_\omega (A_\omega^\top A_\omega)^\dagger A_\omega^\top \in \mathbb{R}^{K \times K}$.
- $R_\omega := I_K - P_\omega \in \mathbb{R}^{K \times K}$, where $I_K$ is the identity matrix of dimension $K$.
- $V_t = A_{N_t}^\top A_{N_t} = A^\top D_{N_t} A = \sum_{k \in [K]} N_t^k \phi_k \phi_k^\top = \sum_{s \le t} \phi_{k_s} \phi_{k_s}^\top$.
- $\widehat{\theta}_t := (A_{N_t}^\top A_{N_t})^\dagger A_{N_t}^\top D_{N_t}^{1/2} \widehat{\mu}_t$, which is the standard least-squares estimator, where $\dagger$ denotes the matrix pseudo-inverse.
- $\tilde{\theta}_t$ and $\tilde{\eta}_t$, parameters for the linear and misspecification parts of the projection $\tilde{\mu}_t$ of empirical mean $\widehat{\mu}_t$ onto set $\mathcal{M}$, such that $\tilde{\mu}_t = A\tilde{\theta}_t + \tilde{\eta}_t$.
- $\theta_t := (A_{N_t}^\top A_{N_t})^\dagger A_{N_t}^\top D_{N_t}^{1/2} \mu$, such that $A\theta_t = D_{N_t}^{-1/2} P_{N_t} D_{N_t}^{1/2} \mu$ if $D_{N_t}$ is invertible. $\theta_t$ is the linear part of the orthogonal parametrization of $\mu$ at time $t$ (see paragraph "Estimation" in Section 4.1 in the main paper).
- $\eta_t := \mu - A\theta_t$, equal to $D_{N_t}^{-1/2} R_{N_t} D_{N_t}^{1/2} \mu$ if $D_{N_t}$ is invertible, is the misspecification part of the orthogonal parametrization of model $\mu$ at time $t$.
- $S_t := D_{N_t}(\widehat{\mu}_t - \mu) \in \mathbb{R}^K$.

## B  The orthogonal parameterization and its properties

Throughout the appendix, we shall adopt an orthogonal parametrization for mean vectors in the model $\mathcal{M}$. In particular, we leverage the following observation: any mean vector $\mu = A\theta + \eta$ can be equivalently represented, at any time $t$, as $\mu = A\theta_t + \eta_t$, where

$$\theta_t := (A_{N_t}^\top A_{N_t})^\dagger A_{N_t}^\top D_{N_t}^{1/2} \mu = V_t^{-1} \sum_{s=1}^{t} \mu^{k_s} \phi_{k_s}$$

is the orthogonal projection (according to the design matrix $V_t$) of $\mu$ onto the feature space and $\eta_t = \mu - A\theta_t$ is the residual. We now introduce some important properties of this parameterization.

**Projecting the empirical mean**  When we use the orthogonal projection described above on the empirical mean $\widehat{\mu}_t$, the resulting linear part is exactly the standard least squares estimator. That is,

$$\widehat{\theta}_t := (A_{N_t}^\top A_{N_t})^\dagger A_{N_t}^\top D_{N_t}^{1/2} \widehat{\mu}_t$$

**Projection matrices**  For $\omega \in \mathbb{R}_{\ge 0}^K$, let us define the projection matrix $P_\omega := A_\omega (A_\omega^\top A_\omega)^\dagger A_\omega^\top \in \mathbb{R}^{K \times K}$ and the residual matrix $R_\omega := I_K - P_\omega \in \mathbb{R}^{K \times K}$. It is easy to check that both are orthogonal projection matrices, i.e., they are symmetric and idempotent ($P_\omega^2 = P_\omega$ and $R_\omega^2 = R_\omega$). Moreover, $P_\omega R_\omega = R_\omega P_\omega = 0$. Equipped with these matrices, we have the following useful identities:

$$A_{N_t} \theta_t = P_{N_t} D_{N_t}^{1/2} \mu = P_{N_t} D_{N_t}^{1/2} A\theta_t,$$

$$D_{N_t}^{1/2} \eta_t = R_{N_t} D_{N_t}^{1/2} \mu = R_{N_t} D_{N_t}^{1/2} \eta_t.$$

**Distances between mean vectors in the model**  Often we will need to compute quantities of the form $\|\lambda - \mu\|_{D_{N_t}}^2$ for different mean vectors in the model. The following lemma shows how to leverage their orthogonal decomposition to split the norm into a distance between their linear parts and a distance between their deviation from linearity.

**Lemma 4** (Linear/non-linear decomposition). *For any $\lambda \in \mathcal{M}$ and $t \ge 1$, there exist $\theta_t' \in \mathbb{R}^d$ and $\eta_t' \in \mathbb{R}^K$ such that $\lambda = A\theta_t' + \eta_t'$ and*

$$\|\lambda - \mu\|_{D_{N_t}}^2 = \|\theta_t' - \theta_t\|_{V_t}^2 + \left\| R_{N_t} D_{N_t}^{1/2} \eta_t' - R_{N_t} D_{N_t}^{1/2} \eta_t \right\|_2^2,$$

$$\|\lambda - \widehat{\mu}_t\|_{D_{N_t}}^2 = \left\| \theta_t' - \widehat{\theta}_t \right\|_{V_t}^2 + \left\| R_{N_t} D_{N_t}^{1/2} \eta_t' - R_{N_t} D_{N_t}^{1/2} \widehat{\mu}_t \right\|_2^2.$$

*Proof.* By leveraging the properties of the orthogonal decomposition and of the matrices $P_{N_t}, R_{N_t}$ (in particular, $P_{N_t} R_{N_t} = 0$ and $P_{N_t} + R_{N_t} = I_K$),

$$\|\lambda - \mu\|_{D_{N_t}}^2 = \|P_{N_t} D_{N_t}(\lambda - \mu) + R_{N_t} D_{N_t}(\lambda - \mu)\|_2^2$$

$$= \left\|P_{N_t} D_{N_t}^{1/2}\lambda - P_{N_t} D_{N_t}^{1/2}\mu\right\|_2^2 + \left\|R_{N_t} D_{N_t}^{1/2}\lambda - R_{N_t} D_{N_t}^{1/2}\mu\right\|_2^2$$

$$= \|P_{N_t} A_{N_t} \theta_t' - P_{N_t} A_{N_t} \theta_t\|_2^2 + \left\|R_{N_t} D_{N_t}^{1/2}\eta_t' - R_{N_t} D_{N_t}^{1/2}\eta_t\right\|_2^2$$

$$= \|\theta_t' - \theta_t\|_{V_t}^2 + \left\|R_{N_t} D_{N_t}^{1/2}\eta_t' - R_{N_t} D_{N_t}^{1/2}\eta_t\right\|_2^2 .$$

The second result can be shown analogously by noting that the projection of $\widehat{\mu}_t$ onto the linear space spanned by $A$ is exactly the least-squares estimator $\widehat{\theta}_t$. □

**The non-linear part of orthogonal parameterizations**  When applying the orthogonal parameterization to a mean vector $\mu = A\theta + \eta$ with $\|\eta\|_\infty \leq \varepsilon$, while we get some crucial properties for the linear part $\theta_t$ (like concentration, see Appendix E), it may be that the resulting non-linear part $\eta_t$ is such that $\|\eta_t\|_\infty > \epsilon$. However, the following result shows that $\eta_t$ cannot be too distant from $\eta$ and, in particular, that $\|\eta_t\|_\infty$ still decreases with $\epsilon$.

**Lemma 5** (Maximum deviation). *Let $t$ any time step such that $V_t$ is invertible. Consider the orthogonal parameterization $(\theta_t, \eta_t)$ for $\mu = A\theta + \eta$ with $\|\eta\|_\infty \leq \varepsilon$. Then,*

$$\|\eta_t\|_\infty \leq (LK + 1)\varepsilon.$$

*Proof.* By definition of the orthogonal parameterization, it is easy to see that $\eta_t - \eta = A(\theta - \theta_t)$. Moreover,

$$\theta_t := (A_{N_t}^\top A_{N_t})^\dagger A_{N_t}^\top D_{N_t}^{1/2}\mu = (A_{N_t}^\top A_{N_t})^\dagger A_{N_t}^\top D_{N_t}^{1/2}(A\theta + \eta)$$

$$= \theta + (A_{N_t}^\top A_{N_t})^\dagger A_{N_t}^\top D_{N_t}^{1/2}\eta = \theta + V_t^{-1} A^\top D_{N_t}\eta = \theta + V_t^{-1}\sum_{k\in[K]} N_t^k \phi_k \eta^k.$$

Therefore, for any arm $k \in [K]$:

$$\left|\eta_t^k - \eta^k\right| = \left|\phi_k^\top V_t^{-1}\sum_{j\in[K]} N_t^j \phi_j \eta^j\right| \overset{(a)}{\leq} \|\phi_k\|_2 \left\|V_t^{-1}\sum_{j\in[K]} N_t^j \phi_j \eta^j\right\|_2$$

$$= \|\phi_k\|_2 \left\|\sum_{j\in[K]} N_t^j \phi_j \eta^j\right\|_{V_t^{-2}} \overset{(b)}{\leq} \|\phi_k\|_2\epsilon \sum_{j\in[K]} N_t^j \|\phi_j\|_{V_t^{-2}} \overset{(c)}{\leq} \|\phi_k\|_2\epsilon K,$$

where (a) is from Cauchy-Schwartz inequality, (b) uses the sub-additivity of the norm, and (c) uses that, for each $j \in [K]$, $V_t = \sum_{q\in[K]} N_t^q \phi_q \phi_q^\top \succeq N_t^j \phi_j \phi_j^\top$ (in the sense of the partial order on positive definite matrices). Using that features are bounded by $L$ in $\ell_2$-norm,

$$\|\eta_t - \eta\|_\infty \leq LK\epsilon,$$

from which the result easily follows. □

**The linear parts of different parametrizations**  We consider mainly two parametrizations of $\mu$: the orthogonal parametrization with respect to $N_t$ and another $(\theta, \eta)$ for which $\|\eta\|_\infty \leq \varepsilon$. We will now relate the linear parts of these two parametrizations.

**Lemma 6.** *Let $t$ any time step such that $V_t$ is invertible. Consider the orthogonal parameterization $(\theta_t, \eta_t)$ for $\mu = A\theta + \eta$ with $\|\eta\|_\infty \leq \varepsilon$. Then*

$$\|\theta_t - \theta\|_{V_t} \leq \sqrt{t}\varepsilon .$$

*Proof.* We use the expression $\theta_t = \theta + V_t^{-1} A^\top D_{N_t} \eta$ derived in the last paragraph, the fact that $P_{N_t}$ is a projection and lastly $\|\eta\|_\infty \le \varepsilon$:

$$\|\theta_t - \theta\|_{V_t} = \left\| V_t^{-1} A^\top D_{N_t} \eta \right\|_{V_t}$$
$$= \sqrt{\eta^\top D_{N_t} A V_t^{-1} A^\top D_{N_t} \eta}$$
$$= \left\| D_{N_t}^{1/2} \eta \right\|_{P_{N_t}} \le \left\| D_{N_t}^{1/2} \eta \right\| = \|\eta\|_{D_{N_t}} \le \sqrt{t}\varepsilon .$$

$\square$

# C Tractable lower bound for the general Top-$m$ identification problem

We present here the proofs for the claims made in the main paper in Section 3.

## C.1 Proof of Lemma 1 and Theorem 1

**Lemma.** (Lemma 1 in the main paper) $\forall \mu, \lambda \in \mathbb{R}^K$ *s.t.* $|S^\star(\mu)| = m$,

$$\mathcal{S}_m(\lambda) \cap \mathcal{S}_m(\mu) = \emptyset \quad \Leftrightarrow \quad \exists i \notin S^\star(\mu) \, \exists j \in S^\star(\mu), \lambda^i > \lambda^j .$$

*Proof.* To see this, first suppose that the condition on the right holds. That is, there exist $(i, j) \in (S^\star(\mu))^c \times S^\star(\mu)$, where $|S^\star(\mu)| = m$, such that $\lambda^i > \lambda^j$. Then, we have two cases. If $j$ does not belong to any of the top-$m$ sets of $\lambda$, that is, $j \notin S^\star(\lambda)$, the result follows trivially since $j$ belongs to the top-$m$ set of $\mu$ $S^\star(\mu)$ and $\mathcal{S}_m(\mu) = \{S^\star(\mu)\}$. If, on the other hand, $j$ belongs to at least one top-$m$ set of $\lambda$, that is, $j \in S^\star(\lambda)$, then $i \in S^\star(\lambda)$ as well since $\lambda^i > \lambda^j$. But $i \notin S^\star(\mu)$, which proves that $\mathcal{S}_m(\lambda) \cap \mathcal{S}_m(\mu) = \emptyset$. Suppose now that $\mathcal{S}_m(\lambda) \cap \mathcal{S}_m(\mu) = \emptyset$ holds and, by contradiction, that $\forall i \notin S^\star(\mu) \, \forall j \in S^\star(\mu), \lambda^i \le \lambda^j$. This trivially implies that $S^\star(\mu)$ is a valid top-$m$ set of $\lambda$. That is, $\mathcal{S}_m(\lambda) \cap \mathcal{S}_m(\mu) \ne \emptyset$ and we have our desired contradiction. $\square$

**Theorem.** (Theorem 1 in the main paper) *For any $\delta \le 1/2$, for any $\delta$-correct algorithm $\mathfrak{A}$ on $\mathcal{M}$, for any bandit instance $\mu \in \mathcal{M}$ such that $|S^\star(\mu)| = m$, the following lower bound holds on the stopping time $\tau_\delta$ of $\mathfrak{A}$ on instance $\mu$:*

$$\mathbb{E}_\mu^{\mathfrak{A}}[\tau_\delta] \ge \left( \sup_{\omega \in \Delta_K} \min_{i \notin S^\star(\mu)} \min_{j \in S^\star(\mu)} \inf_{\lambda \in \mathcal{M}: \lambda^i > \lambda^j} \sum_{k \in [K]} \omega^k \mathrm{KL}(\mu^k, \lambda^k) \right)^{-1} \log\left( \frac{1}{2.4\delta} \right) .$$

*Proof.* We start from Equation 2 (main paper), and using Lemma 1, we can rewrite the inf operator. That yields the desired expression. $\square$

## C.2 Proof of Lemma 2

Let $\Lambda_m(\mu, \mathcal{M}') \subseteq \mathcal{M}'$ denote the set of alternative models to $\mu \in \mathbb{R}^K$ in the model $\mathcal{M}'$ (which might be different from $\mathcal{M}$). Consider the lower bound problem

$$H_\mu(\mathcal{M}') := \sup_{\omega \in \Delta_K} \inf_{\lambda \in \Lambda_m(\mu, \mathcal{M}')} \sum_{k \in [K]} \omega^k \, \mathrm{KL}(\mu^k, \lambda^k) .$$

A pair of equilibrium strategies for that problem is composed of $\omega \in \Delta_K$ and $q \in \mathcal{P}(\Lambda_m(\mu, \mathcal{M}'))$ (which is the set of probability distributions on $\Lambda_m(\mu, \mathcal{M}')$). Let $Q_{\mathcal{M}'}$ be the set of equilibrium distributions. For $q \in Q_{\mathcal{M}'}$, let $\Lambda_q \subseteq \Lambda_m(\mu, \mathcal{M}')$ be its support.

**Lemma 7.** *Let $\mathcal{M}_1, \mathcal{M}_2$ be models such that $\mathcal{M}_1 \subseteq \mathcal{M}_2$. For any $q \in Q_{\mathcal{M}_2}$, if $\Lambda_q \subseteq \mathcal{M}_1$, then $H_\mu(\mathcal{M}_1) = H_\mu(\mathcal{M}_2)$.*

*Proof.* First, we have $H_\mu(\mathcal{M}_1) \geq H_\mu(\mathcal{M}_2)$ since $\mathcal{M}_1 \subseteq \mathcal{M}_2$. If $\Lambda_q \subseteq \mathcal{M}_1$, then using successively $q \in \mathcal{P}(\Lambda(\mu, \mathcal{M}_1))$ and $q \in Q_{\mathcal{M}_2}$,

$$H_\mu(\mathcal{M}_1) = \sup_{\omega \in \Delta_K} \inf_{\lambda \in \Lambda_m(\mu, \mathcal{M}_1)} \sum_{k \in [K]} \omega^k \, \mathrm{KL}(\mu^k, \lambda^k)$$

$$\leq \sup_{\omega \in \Delta_K} \mathbb{E}_{\lambda \sim q} \sum_{k \in [K]} \omega^k \, \mathrm{KL}(\mu^k, \lambda^k) = H_\mu(\mathcal{M}_2) \,.$$

$\square$

For $\lambda \in \mathbb{R}^K$, let $|\lambda|_\varepsilon = \inf\{\|\eta\|_\infty \mid \exists \theta \in \mathbb{R}^d, \ \lambda = A\theta + \eta\}$. Let us now consider $\mathcal{M}$ as defined in Equation 1 in the main paper, with misspecification upper bound $\varepsilon \geq 0$.

**Lemma 8.** *Let $\mathcal{M}' \subseteq \{\lambda \in \mathbb{R}^K \mid \|\lambda\|_\infty \leq M\}$ be a set of models such that $\mathcal{M} \subseteq \mathcal{M}'$ and $\varepsilon > \varepsilon_\mu(\mathcal{M}') := \inf_{q \in Q_{\mathcal{M}'}} \sup_{\lambda \in \Lambda_q} |\lambda|_\varepsilon$.[6] Then $H_\mu(\mathcal{M}) = H_\mu(\mathcal{M}')$.*

*Proof.* If $\varepsilon > \inf_{q \in Q_{\mathcal{M}'}} \sup_{\lambda \in \Lambda_q} |\lambda|_\varepsilon$, then there exists $q \in Q_{\mathcal{M}'}$ such that for all $\lambda \in \Lambda_q$, $|\lambda|_\varepsilon \leq \varepsilon$. Hence $\Lambda_q \subseteq \mathcal{M}$ and we apply Lemma 7. $\square$

For any model $\mathcal{M}'$, there exist equilibrium strategies for which $q$ is supported on $K$ points [11]. Hence $\varepsilon_\mu(\mathcal{M}')$ is always finite.

Let $\mathcal{M}_u := \mathbb{R}^K$ be the set of unstructured models, and for $a, b \in \mathbb{R}$, $\mathcal{M}_{[a,b]} := \{\lambda \in \mathbb{R}^K \mid \forall k \in [K], \lambda^k \in [a,b]\}$ be the set of models that verify a boundedness assumption.

**Lemma 9.** *Let $\mu^{(K)} := \min_j \mu^j$ and $\mu^{(1)} := \max_j \mu^j$. For all $\mu \in \mathbb{R}^K$, $H_\mu(\mathcal{M}_u) = H_\mu(\mathcal{M}_{[\mu^{(K)}, \mu^{(1)}]})$.*

*Proof.* Let us consider any $\lambda \in \Lambda_m(\mu, \mathcal{M}_u)$, such that there exists $k \in [K]$ with $\lambda^k \notin [\mu^{(K)}, \mu^{(1)}]$. Let us define $\tilde{\lambda}$ as the projection of $\lambda$ onto $[\mu^{(K)}, \mu^{(1)}]^K$. Then $\tilde{\lambda}$ satisfies $\tilde{\lambda} \in \Lambda_m(\mu, \mathcal{M}_{[\mu^{(K)}, \mu^{(1)}]}) \subseteq \Lambda_m(\mu, \mathcal{M}_u)$, and by monotonicity of the Kullback-Leibler divergence in one-parameter exponential families, for all $k \in [K]$, $\mathrm{KL}(\mu^k, \tilde{\lambda}^k) \leq \mathrm{KL}(\mu^k, \lambda^k)$. Thus for all $\omega \in \Delta_K$

$$\sum_{k \in [K]} \omega^k \, \mathrm{KL}(\mu^k, \tilde{\lambda}^k) \leq \sum_{k \in [K]} \omega^k \, \mathrm{KL}(\mu^k, \lambda^k) \,.$$

For $q \in Q_{\mathcal{M}_u}$, let $\tilde{q}$ be the distribution in which every support point $\lambda$ of $q$ is transported onto its projection $\tilde{\lambda}$. Then for all $\omega \in \triangle_K$,

$$\mathbb{E}_{\lambda \sim \tilde{q}} \sum_{k \in [K]} \omega^k \, \mathrm{KL}(\mu^k, \lambda^k) \leq \mathbb{E}_{\lambda \sim q} \sum_{k \in [K]} \omega^k \, \mathrm{KL}(\mu^k, \lambda^k) \,,$$

from which we obtain that $\tilde{q}$ has lower objective value than $q$. Since $q \in Q_{\mathcal{M}_u}$, then $\tilde{q} \in Q_{\mathcal{M}_u}$ as well. By construction, its support verifies $\Lambda_{\tilde{q}} \subseteq \mathcal{M}_{[\mu^{(K)}, \mu^{(1)}]}$. We conclude with Lemma 7. $\square$

Applying Lemma 8 to $\mathcal{M}_{[\mu^{(K)}, \mu^{(1)}]}$, together with Lemma 9, we finally obtain Lemma 2 from the main paper, restated here using the notations we introduced:

**Lemma.** *If $\varepsilon > \mu^{(1)} - \mu^{(K)}$ then $H_\mu(\mathcal{M}) = H_\mu(\mathcal{M}_{[\mu^{(K)}, \mu^{(1)}]}) = H_\mu(\mathcal{M}_u)$.*

### C.3 Computing the closest alternative

In order to compute the closest alternative to $\mu \in \mathcal{M}$ in the half-space $\{\lambda \in \mathcal{M} \mid \lambda^k \geq \lambda^j\}$, the optimization problem we need to solve is

$$\inf_{\theta, \eta} \frac{1}{2} \|A\theta + \eta - \mu\|^2_{D_{N_t}}$$

$$\text{s.t } (e_k - e_j)^\top (A\theta + \eta) \geq 0$$

$$\|A\theta + \eta\|_\infty \leq M$$

$$\|\eta\|_\infty \leq \varepsilon \,.$$

---

[6]Note that indeed quantity $\varepsilon_\mu(\mathcal{M}')$ depends on $\mu$, since $Q_{\mathcal{M}'}$ is defined with respect to $\mu$.

In our implementation, and thus in the remainder of this section, we shall drop the boundedness constraint $\|A\theta + \eta\|_\infty \leq M$ which has typically a negligible effect on the algorithm's behavior.

**Quadratic problem**  We express the problem as function of the variable $(\theta^\top, \eta^\top)^\top$. Up to the constant term, this problem is equivalent to

$$\inf_{\theta,\eta} \begin{pmatrix} \theta \\ \eta \end{pmatrix}^\top \begin{pmatrix} A^\top D_N A & A^\top D_N \\ D_N A & D_N \end{pmatrix} \begin{pmatrix} \theta \\ \eta \end{pmatrix} - \begin{pmatrix} \theta \\ \eta \end{pmatrix}^\top \begin{pmatrix} A^\top D_N \mu \\ D_N \mu \end{pmatrix}$$

$$\text{s.t.} \begin{pmatrix} A^\top(e_j - e_k) \\ e_j - e_k \end{pmatrix}^\top \begin{pmatrix} \theta \\ \eta \end{pmatrix} \leq 0$$

$$\|\eta\|_\infty \leq \varepsilon.$$

In the code, we directly solve the problem under this form using a quadratic problem solver.

**Computing the closest alternative**  We now detail the form of the solutions analytically (as much as possible). Let $j, k \in [K]$, $j \neq k$. We want to compute the closest alternative in the half-space $\{\lambda \in \mathcal{M} \mid \lambda^k \geq \lambda^j\}$ to $\mu \in \mathbb{R}^K$. That is, we compute the solution to

$$\inf_{\theta,\eta} \frac{1}{2} \|A\theta + \eta - \mu\|^2_{D_{N_t}}$$

$$\text{s.t } (e_k - e_j)^\top (A\theta + \eta) \geq 0$$

$$\eta \in \mathcal{C}$$

Here, to highlight the generality of the following derivation, we replace the $\ell_\infty$ norm constraint on $\eta$ with any convex set $\mathcal{C}$. To simplify the notation, we denote by $D_N$ the diagonal matrix with $N_t$ on the diagonal and $u := e_j - e_k$. The problem above is then written as

$$\inf_{\theta,\eta} \frac{1}{2} \left\| D_N^{1/2} A\theta + D_N^{1/2}\eta - D_N^{1/2}\mu \right\|^2_2$$

$$\text{s.t } u^\top (A\theta + \eta) \leq 0$$

$$\eta \in \mathcal{C}$$

**Assumption 1.** At $t_0$, $A^\top D_{N_{t_0}} A = V_{t_0}$ is invertible.

See paragraph "Initialization phase" in Subsection 4.1 to see how that assumption is ensured in practice. We now suppose that $t \geq t_0$. Minimizing first in $\theta$ at fixed $\eta$, we solve the problem

$$\inf_{\theta} \frac{1}{2} \left\| D_N^{1/2} A\theta + D_N^{1/2}\eta - D_N^{1/2}\mu \right\|^2_2$$

$$\text{s.t } u^\top (A\theta + \eta) \leq 0$$

The Lagrangian is $L(\theta, \alpha) = \frac{1}{2} \left\| D_N^{1/2} A\theta + D_N^{1/2}\eta - D_N^{1/2}\mu \right\|^2_2 + \alpha u^\top (\eta + A\theta)$ with $\alpha \geq 0$. We get that at the optimal $\theta$,

$$A^\top D_N(A\theta + \eta - \mu) = -\alpha A^\top u \quad \implies \quad \theta = (A^\top D_N A)^{-1} A^\top (-\alpha u + D_N \mu - D_N \eta).$$

At the optimum, from the KKT conditions, either $\alpha = 0$ and $u^\top (A\theta + \eta) \leq 0$, or $\alpha > 0$ and $u^\top A\theta = -u^\top \eta$.

**Case $\alpha = 0$.**  If $\alpha = 0$, then $\theta = (A^\top D_N A)^{-1} A^\top D_N (\mu - \eta)$, $D_N^{1/2}(A\theta + \eta - \mu) = (D_N^{1/2} A (A^\top D_N A)^{-1} A^\top D_N^{1/2} - I) D_N^{1/2}(\mu - \eta)$ and the value of the optimization problem is the norm of this quantity.

Let $P_N = D_N^{1/2} A (A^\top D_N A)^{-1} A^\top D_N^{1/2}$. Note: it is symmetric and idempotent ($P_N^2 = P_N$), meaning that it is an orthogonal projection. Let $R_N = I - P_N$ be the residual matrix. We also have $R_N^2 = R_N$. Furthermore, $P_N R_N = R_N P_N = 0$.

With these notations, $D_N^{1/2} A\theta = P_N D_N^{1/2}(\mu - \eta)$, $D_N^{1/2}(A\theta + \eta - \mu) = -R_N D_N^{1/2}(\mu - \eta)$ and the value of the optimization problem is $\frac{1}{2} \|R_N D_N^{1/2}\eta - R_N D_N^{1/2}\mu\|^2$. The case $\alpha = 0$ is

possible only if the constraint is then satisfied, that is if $u^\top(A\theta + \eta) \leq 0$ at the optimum, i.e. if $u^\top(A^\top(A^\top D_N A)^{-1}A^\top D_N\mu + (I - A^\top(A^\top D_N A)^{-1}A^\top D_N)\eta) \leq 0$. The problem we need to solve in that case is

$$\min_{\eta_N} \quad \frac{1}{2}\left\|R_N D_N^{1/2}\eta - R_N D_N^{1/2}\mu\right\|_2^2$$

$$\text{s.t. } u^\top(I - A^\top(A^\top D_N A)^{-1}A^\top D_N)\eta \leq -u^\top A^\top(A^\top D_N A)^{-1}A^\top D_N\mu$$

$$\eta \in \mathcal{C}$$

If $\mathcal{C}$ is convex this is a convex optimization problem. It can happen that there is no feasible point, which simply means that there is no solution with $\alpha = 0$.

**Case $\alpha \neq 0$.** Consider now the case $\alpha > 0$. We get

$$u^\top A\theta = -u^\top\eta$$

$$\implies u^\top A(A^\top D_N A)^{-1}A^\top(-\alpha u + D_N\mu - D_N\eta) = -u^\top\eta$$

$$\implies \alpha u^\top A(A^\top D_N A)^{-1}A^\top u = u^\top A(A^\top D_N A)^{-1}A^\top D_N(\mu - \eta) + u^\top\eta$$

Then

$$D_N^{1/2}A\theta = D_N^{1/2}A(A^\top D_N A)^{-1}A^\top(-\alpha u + D_N\mu - D_N\eta)$$

$$= -\alpha D_N^{1/2}A(A^\top D_N A)^{-1}A^\top u + P_N D_N^{1/2}(\mu - \eta)$$

$$D_N^{1/2}(A\theta + \eta - \mu) = -\alpha D_N^{1/2}A(A^\top D_N A)^{-1}A^\top u - R_N D_N^{1/2}(\mu - \eta)$$

$$= -\frac{u^\top A(A^\top D_N A)^{-1}A^\top D_N(\mu - \eta) + u^\top\eta}{u^\top A(A^\top D_N A)^{-1}A^\top u}D_N^{1/2}A(A^\top D_N A)^{-1}A^\top u$$

$$- R_N D_N^{1/2}(\mu - \eta)$$

We can now see that $D_N^{1/2}(A\theta + \eta - \mu)$ is linear in $\eta$ and the objective value $\frac{1}{2}\left\|D_N^{1/2}(A\theta + \eta - \mu)\right\|^2$ is quadratic in $\eta$. We need to solve a quadratic optimization problem under the constraint $\eta \in \mathcal{C}$. Let's now simplify that optimization problem. We first show that the cross term in $\frac{1}{2}\left\|D_N^{1/2}(A\theta + \eta - \mu)\right\|_2^2 = \frac{1}{2}\left\|-\alpha D_N^{1/2}A(A^\top D_N A)^{-1}A^\top u - R_N D_N^{1/2}(\mu - \eta)\right\|_2^2$ is zero. Note: if $D_N$ is invertible, then $\frac{1}{2}\left\|-\alpha D_N^{1/2}A(A^\top D_N A)^{-1}A^\top u - R_N D_N^{1/2}(\mu - \eta)\right\|_2^2 = \frac{1}{2}\left\|-\alpha P_N D_N^{-1/2}u - R_N D_N^{1/2}(\mu - \eta)\right\|_2^2$ and the fact that the cross term is 0 is a simple consequence of $P_N R_N = R_N P_N = 0$.

$$(R_N D_N^{1/2}(\mu - \eta))^\top D_N^{1/2}A(A^\top D_N A)^{-1}A^\top u$$

$$= ((I - P_N)D_N^{1/2}(\mu - \eta))^\top D_N^{1/2}A(A^\top D_N A)^{-1}A^\top u$$

$$= (\mu - \eta)^\top D_N A(A^\top D_N A)^{-1}A^\top u - (\mu - \eta)^\top D_N^{1/2}P_N D_N^{1/2}A(A^\top D_N A)^{-1}A^\top u$$

$$= (\mu - \eta)^\top D_N A(A^\top D_N A)^{-1}A^\top u - (\mu - \eta)^\top D_N A(A^\top D_N A)^{-1}A^\top D_N A(A^\top D_N A)^{-1}A^\top u$$

$$= 0.$$

Now that we established that the cross term is zero, the objective value is simply the sum of two square terms,

$$\frac{1}{2}\left\|D_N^{1/2}(A\theta + \eta - \mu)\right\|_2^2 = \frac{1}{2}\alpha^2 u^\top A(A^\top D_N A)^{-1}A^\top u + \frac{1}{2}(\mu - \eta)^\top D_N^{1/2}R_N D_N^{1/2}(\mu - \eta)$$

$$= \frac{1}{2}\frac{\left(u^\top A(A^\top D_N A)^{-1}A^\top D_N(\mu - \eta) + u^\top\eta\right)^2}{u^\top A(A^\top D_N A)^{-1}A^\top u}$$

$$+ \frac{1}{2}(\mu - \eta)^\top D_N^{1/2}R_N D_N^{1/2}(\mu - \eta)$$

$$= \frac{1}{2}\eta^\top Q\eta + q^\top\eta + C$$

where $C$ doesn't depend on $\eta$ and

$$
\begin{aligned}
Q = {} & D_N^{1/2} R_N D_N^{1/2} \\
& + \frac{1}{u^\top A(A^\top D_N A)^{-1} A^\top u} \left( (I - D_N A(A^\top D_N A)^{-1} A^\top) u \right) \left( (I - D_N A(A^\top D_N A)^{-1} A^\top) u \right)^\top
\end{aligned}
$$

$$
q = \frac{u^\top A(A^\top D_N A)^{-1} A^\top D_N \mu}{u^\top A(A^\top D_N A)^{-1} A^\top u} (I - D_N A(A^\top D_N A)^{-1} A^\top) u - D_N^{1/2} R_N D_N^{1/2} \mu \, .
$$

Again if $D_N$ is invertible these have simpler expressions:

$$
Q = D_N^{1/2} \left( R_N + \frac{1}{u^\top D_N^{-1/2} P_N D_N^{-1/2} u} (R_N D_N^{-1/2} u)(R_N D_N^{-1/2} u)^\top \right) D_N^{1/2}
$$

$$
q = D_N^{1/2} R_N \left( \frac{u^\top D_N^{-1/2} P_N D_N^{1/2} \mu}{u^\top D_N^{-1/2} P_N D_N^{-1/2} u} D_N^{-1/2} u - D_N^{1/2} \mu \right) \, .
$$

We are looking for a solution to

$$
\arg \min_{\eta \in \mathcal{C}} \frac{1}{2} \eta^\top Q \eta + q^\top \eta \, .
$$

This is a quadratic objective. The difficulty of finding the minimum depends on $\mathcal{C}$.

**Summary.** To compute the closest alternative in a half-space, we compute the solution to two quadratic problems corresponding to the possibilities that Lagrangian multiplier $\alpha$ satisfies either $\alpha = 0$ or $\alpha > 0$. Then we retain the solution with the minimal objective value.

# D  The MISLID algorithm

## D.1  Initialization

MISLID starts by pulling a deterministic sequence of $t_0$ arms that make the minimum eigenvalue of the resulting design matrix $V_{t_0}$ larger than $2L^2$. Since the rows of $A$ span $\mathbb{R}^d$, such sequence can be found by taking any subset of $d$ arms that span the whole space (e.g., by computing a barycentric spanner [4]) and pulling them in a round robin fashion until the desired condition is met.

In order to get an approximation of the length $t_0$ of the initialization phase, let us denote $\sigma_{\min}(M)$ the minimal singular value of a matrix $M$. Let us consider $\mathcal{B} = \{b_1, b_2, \ldots, b_d\} \subseteq [K]$, $|\mathcal{B}| = d$, the barycentric spanner of size $d$ computed on matrix $A$. Then, if we stopped the round-robin sampling such that each arm in the barycentric spanner is sampled exactly $u_0$ times, $V_{t_0} = u_0 \sum_{k \in \mathcal{B}} \phi_k \phi_k^\top$. To ensure that $V_t \succeq 2L^2 I_d$, we need $u_0 \sigma_{\min} \left( \sum_{k \in \mathcal{B}} \phi_k \phi_k^\top \right) \geq 2L^2$. Let $\Gamma'(A) := \min \left\{ \sigma_{\min} \left( \sum_{k \in \mathcal{B}} \phi_k \phi_k^\top \right) \mid \mathcal{B} \ d\text{-sized spanner of } A \right\}$. Then $u_0 = \left\lceil \frac{2L^2}{\Gamma'(A)} \right\rceil$ is large enough.

We obtain the bound $t_0 \leq d \left\lceil \frac{2L^2}{\Gamma'(A)} \right\rceil$.

## D.2  Projection of the empirical mean onto the set of realizable models $\mathcal{M}$

As done in Equation 1 in the main paper, we define the set of realizable models as

$$
\mathcal{M} := \left\{ \mu = A\theta + \eta \in \mathbb{R}^K \mid \exists \theta \in \mathbb{R}^d \exists \eta \in \mathbb{R}^K, \|\eta\|_\infty \leq \varepsilon \wedge \|A\theta + \eta\|_\infty \leq M \right\} \, .
$$

We require our estimates of $\mu$ to be in this set, but the estimate at time $t$ $\widehat{\mu}_t$ might not satisfy the constraint on its $\ell_\infty$ norm (*i.e.*, $\|\widehat{\mu}_t\|_\infty > M$). We then directly project the empirical mean vector onto $\mathcal{M}$. Define

$$
(\tilde{\theta}_t, \tilde{\eta}_t) := \underset{\theta', \eta' : A\theta' + \eta' \in \mathcal{M}}{\arg \min} \|A\theta' + \eta' - \widehat{\mu}_t\|_{D_{N_t}}^2 \, . \tag{6}
$$

**Lemma 10.** *Let* $\tilde{\mu}_t = A\tilde{\theta}_t + \tilde{\eta}_t,$[7] *where* $(\tilde{\theta}_t, \tilde{\eta}_t)$ *are the solution of* (6)*. Then, all the following hold:*

$$\|\tilde{\mu}_t - \mu\|_{D_{N_t}}^2 \leq \|\mu - \widehat{\mu}_t\|_{D_{N_t}}^2 \;,$$

$$\left\|\tilde{\theta}_t - \theta_t\right\|_{V_t}^2 \leq \left\|\widehat{\theta}_t - \theta_t\right\|_{V_t}^2 \;,$$

$$\left\|R_{N_t} D_{N_t}^{1/2} \tilde{\eta}_t - R_{N_t} D_{N_t}^{1/2} \eta_t\right\|_2^2 \leq \left\|R_{N_t} D_{N_t}^{1/2} \widehat{\mu}_t - R_{N_t} D_{N_t}^{1/2} \eta_t\right\|_2^2 \;,$$

$$\left\|\tilde{\theta}_t - \theta\right\|_{V_t}^2 \leq \left\|\widehat{\theta}_t - \theta\right\|_{V_t}^2 \;,$$

$$\left\|R_{N_t} D_{N_t}^{1/2} \tilde{\eta}_t - R_{N_t} D_{N_t}^{1/2} \eta\right\|_2^2 \leq \left\|R_{N_t} D_{N_t}^{1/2} \widehat{\mu}_t - R_{N_t} D_{N_t}^{1/2} \eta\right\|_2^2$$

*Proof.* The first inequality is easy to check by using $\mu \in \mathcal{M}$ together with the non-expansion of the projection in the optimized norm.

The proof of the other inequalities extends Lemma 9 in [40]. Note that, using Lemma 4, an equivalent formulation of (6) is

$$(\tilde{\theta}_t, \tilde{\eta}_t) := \underset{\theta', \eta': A\theta'+\eta' \in \mathcal{M}}{\arg\min} \left\{ \left\|P_{N_t} A_{N_t} \theta' - P_{N_t} D_{N_t}^{1/2} \widehat{\mu}_t\right\|_2^2 + \left\|R_{N_t} D_{N_t}^{1/2} \eta' - R_{N_t} D_{N_t}^{1/2} \widehat{\mu}_t\right\|_2^2 \right\}$$

$$= \underset{\theta', \eta': A\theta'+\eta' \in \mathcal{M}}{\arg\min} \left\{ \left\|\theta' - \widehat{\theta}_t\right\|_{V_t}^2 + \|\eta' - \widehat{\mu}_t\|_{D_{N_t}^{1/2} R_{N_t} D_{N_t}^{1/2}}^2 \right\}$$

This is the minimization of a convex function over a convex set. For any $\theta' \in \mathbb{R}^d$, $\eta' \in \mathbb{R}^K$, let $f(\theta') = \left\|\theta' - \widehat{\theta}_t\right\|_{V_t}^2$ and $g(\eta') = \|\eta' - \widehat{\mu}_t\|_{D_{N_t}^{1/2} R_{N_t} D_{N_t}^{1/2}}^2$. Therefore, using the first-order optimality conditions for convex functions (see, e.g., Theorem 2.8 in [32]), $(\tilde{\theta}_t, \tilde{\eta}_t)$ are minimizers if and only if for each $\theta', \eta' : A\theta' + \eta' \in \mathcal{M}$,

$$\langle \nabla_\theta f(\tilde{\theta}_t), \theta' - \tilde{\theta}_t \rangle \geq 0 \quad \Longrightarrow \quad (\tilde{\theta}_t - \widehat{\theta}_t)^T V_t(\theta' - \tilde{\theta}_t) \geq 0$$

$$\langle \nabla_\eta g(\tilde{\eta}_t), \eta' - \tilde{\eta}_t \rangle \geq 0 \quad \Longrightarrow \quad (\tilde{\eta}_t - \widehat{\mu}_t)^T D_{N_t}^{1/2} R_{N_t} D_{N_t}^{1/2}(\eta' - \tilde{\eta}_t) \geq 0$$

Note that $\mu = A\theta_t + \eta_t$, thus the orthogonal parametrization $(\theta_t, \eta_t)$ is such that $A\theta_t + \eta_t \in \mathcal{M}$. Thus, $(\theta_t, \eta_t)$ are feasible solutions. This implies

$$\left\|\widehat{\theta}_t - \theta_t\right\|_{V_t}^2 = \left\|\widehat{\theta}_t - \tilde{\theta}_t\right\|_{V_t}^2 + \left\|\tilde{\theta}_t - \theta_t\right\|_{V_t}^2 + 2(\widehat{\theta}_t - \tilde{\theta}_t)^T V_t(\tilde{\theta}_t - \theta_t) \geq \left\|\tilde{\theta}_t - \theta_t\right\|_{V_t}^2$$

and

$$\|\widehat{\mu}_t - \eta_t\|_{D_{N_t}^{1/2} R_{N_t} D_{N_t}^{1/2}}^2 = \|\widehat{\mu}_t - \tilde{\eta}_t\|_{D_{N_t}^{1/2} R_{N_t} D_{N_t}^{1/2}}^2 + \|\tilde{\eta}_t - \eta_t\|_{D_{N_t}^{1/2} R_{N_t} D_{N_t}^{1/2}}^2$$

$$+ 2(\widehat{\mu}_t - \tilde{\eta}_t)^T D_{N_t}^{1/2} R_{N_t} D_{N_t}^{1/2}(\tilde{\eta}_t - \eta_t)$$

$$\geq \|\tilde{\eta}_t - \eta_t\|_{D_{N_t}^{1/2} R_{N_t} D_{N_t}^{1/2}}^2 \;.$$

Rearranging concludes the proof of the second and the third inequalities. To show the last two inequalities, simply use the same argument by noting that $(\theta, \eta)$ is also a feasible solution (since $\mu = A\theta + \eta \in \mathcal{M}$). $\qquad\square$

# E Concentration results

## E.1 Concentration of the linear part

In this section we derive concentration results for

---

[7] Note that the equation $\widehat{\theta}_t = \tilde{\theta}_t$ mentioned in Section 4.1 in the main paper no longer holds, because we consider the boundedness assumption $\mu \in \mathcal{M} \implies \|\mu\|_\infty \leq M$

$$\left\|\widehat{\theta}_t - \theta_t\right\|_{V_t}^2 = \left\|V_t^{-1} A^\top S_t\right\|_{V_t}^2 = \left\|A^\top S_t\right\|_{V_t^{-1}}^2 .$$

We rewrite the quantities involved to make obvious that this is the usual self-normalized quantity from the linear bandit literature [1]:

$$A^\top S_t = \sum_{s=1}^t \left(X_s^{k_s} - \mu^{k_s}\right) A^\top e_{k_s} = \sum_{s=1}^t \left(X_s^{k_s} - \mu^{k_s}\right) \phi_{k_s} \quad \text{and} \quad V_t = \sum_{s=1}^t \phi_{k_s} \phi_{k_s}^\top .$$

We restate here Theorem 20.4 (in combination with the Equation 20.9) of [28], which states a result due to [1].

**Theorem 3.** *Suppose that for all $k \in [K]$, $\|\phi_k\|_2 \le L$. For all $x > 0$ and $\delta \in (0, 1]$,*

$$\mathbb{P}\left(\exists t \in \mathbb{N}, \frac{1}{2}\left\|A^\top S_t\right\|_{(V_t + x I_d)^{-1}}^2 \ge \log\frac{1}{\delta} + \frac{d}{2}\log\left(1 + \frac{tL^2}{xd}\right)\right) \le \delta .$$

**Corollary 1.** *If we ensure that $V_{t_0} \succeq x I_d$ (in the sense of positive definite matrices), then*

$$\mathbb{P}\left(\exists t > t_0, \frac{1}{2}\left\|\widehat{\theta}_t - \theta_t\right\|_{V_t}^2 \ge 2\log\frac{1}{\delta} + d\log\left(1 + \frac{tL^2}{xd}\right)\right) \le \delta .$$

*Proof.* If $V_t \succeq x I_d$ then $2V_t \succeq V_t + x I_d$ and

$$\left\|\widehat{\theta}_t - \theta_t\right\|_{V_t}^2 = \left\|A^\top S_t\right\|_{V_t^{-1}}^2 \le 2\left\|A^\top S_t\right\|_{(V_t + x I_d)^{-1}}^2 .$$

$\square$

The $2\log(1/\delta)$ term is fine for some steps of the analysis but not for the stopping rule. For the stopping rule concentration inequality, we need $\log(1/\delta)$.

**Corollary 2.** *Suppose that $V_{t_0} \succeq x I_d$. Then*

$$\mathbb{P}\left(\exists t \ge t_0, \frac{1}{2}\left\|\widehat{\theta}_t - \theta_t\right\|_{V_t}^2 \ge 1 + \log\frac{1}{\delta} + \left(1 + \frac{1}{\log(1/\delta)}\right)\frac{d}{2}\log\left(1 + \frac{tL^2}{xd}\log\frac{1}{\delta}\right)\right) \le \delta .$$

*Proof.* Suppose that $V_{t_0} \succeq x I_d$ and let $\gamma(\delta) := \log(1/\delta)^{-1}$. For any $t \ge t_0$,

$$\left\|\widehat{\theta}_t - \theta_t\right\|_{V_t}^2 = \left\|A^\top S_t\right\|_{V_t^{-1}}^2 \le (1 + \gamma(\delta))\left\|A^\top S_t\right\|_{(V_t + x\gamma(\delta) I_d)^{-1}}^2 .$$

Then we conclude by applying Theorem 3. $\square$

### E.2 Unstructured concentration

Let $W_{-1}$ be the negative branch of the Lambert W function and let $\overline{W}(x) = -W_{-1}(-e^{-x})$. Note that for $x \ge 1$, $x + \log x \le \overline{W}(x) \le x + \log x + \min\{\frac{1}{2}, \frac{1}{\sqrt{x}}\}$.

**Lemma 11.** *For $t > 1$, with probability $1 - \delta$,*

$$\frac{1}{2}\left\|\widehat{\mu}_t - \mu\right\|_{D_{N_t}}^2 \le 2K\overline{W}\left(\frac{1}{2K}\log\frac{e}{\delta} + \frac{1}{2}\log(8eK\log t)\right) .$$

*Proof.* See [14, Appendix A, Theorem 4] for that form of the lemma, which is a small reformulation of a result due to [30]. $\square$

The concentration inequality of Lemma 11 is also valid for $\|\tilde{\mu}_t - \mu\|_{D_{N_t}}^2$ since the first inequality of Lemma 10 states that $\|\tilde{\mu}_t - \mu\|_{D_{N_t}}^2 \le \|\widehat{\mu}_t - \mu\|_{D_{N_t}}^2$.

### E.3 Elliptic potential lemmas

All lemmas in this section are derived under the following assumption.

**Assumption 2.** For $t \geq t_0$, $V_t \succeq 2L^2 I_d$.

In the remainder of the section, we consider $\omega_t \in \Delta^K$, for any time $t > 0$.

**Lemma 12.** *Under Assumption 2, with probability $1 - \delta$,*

$$\sum_{s=t_0+1}^{t} \sum_{k=1}^{K} \omega_s^k \|\phi_k\|_{V_{s-1}^{-1}}^2 \leq \sqrt{2t \log \frac{1}{\delta}} + d \log \left(1 + \frac{t}{d}\right) .$$

*Proof.*

$$\sum_{s=t_0+1}^{t} \sum_{k=1}^{K} \omega_s^k \|\phi_k\|_{V_{s-1}^{-1}}^2 = \sum_{s=t_0+1}^{t} \left( \sum_{k=1}^{K} \omega_s^k \|\phi_k\|_{V_{s-1}^{-1}}^2 - \|\phi_{k_s}\|_{V_{s-1}^{-1}}^2 \right) + \sum_{s=t_0+1}^{t} \|\phi_{k_s}\|_{V_{s-1}^{-1}}^2 .$$

The first term is the sum of a martingale difference sequence with bounded increments

$$\mathbb{E}\left[ \sum_{k=1}^{K} \omega_s^k \|\phi_k\|_{V_{s-1}^{-1}}^2 - \|\phi_{k_s}\|_{V_{s-1}^{-1}}^2 \mid \mathcal{F}_{s-1} \right] = 0 ,$$

$$\left| \sum_{k=1}^{K} \omega_s^k \|\phi_k\|_{V_{s-1}^{-1}}^2 - \|\phi_{k_s}\|_{V_{s-1}^{-1}}^2 \right| \leq 1 .$$

since $V_{s-1} \succeq 2L^2 I_d$ and $\|\phi_k\| \leq L$. By the Azuma-Hoeffding inequality, with probability $1 - \delta$,

$$\sum_{s=t_0+1}^{t} \left( \sum_{k=1}^{K} \omega_s^k \|\phi_k\|_{V_{s-1}^{-1}}^2 - \|\phi_{k_s}\|_{V_{s-1}^{-1}}^2 \right) \leq \sqrt{2t \log \frac{1}{\delta}} .$$

The second term is an elliptic potential, bounded in Lemma 13 below. $\qquad \square$

**Lemma 13.** *Under Assumption 2, for $t > t_0$,*

$$\sum_{s=t_0+1}^{t} \|\phi_{k_s}\|_{V_{s-1}^{-1}}^2 \leq d \log \left(1 + \frac{t}{d}\right) .$$

*Proof.* Let $V_{s:t}$ denote the design matrix using only rounds from $s$ to $t$. We use Lemma 14,

$$\sum_{s=t_0+1}^{t} \|\phi_{k_s}\|_{V_{s-1}^{-1}}^2 \leq \sum_{s=t_0+1}^{t} \|\phi_{k_s}\|_{(2L^2 I_d + V_{t_0+1:s-1})^{-1}}^2 \leq d \log \left(1 + \frac{t}{d}\right) .$$

$\qquad \square$

**Lemma 14.** *Under Assumption 2, for $t > t_0$,*

$$\sum_{s=t_0+1}^{t} \|\phi_{k_s}\|_{(V_{t_0+1:s-1} + 2L^2 I_d)^{-1}}^2 \leq d \log \left(1 + \frac{t}{d}\right) .$$

*Proof.* By definition of $L$, for all $k \in [K]$, $\phi_k \phi_k^\top \preceq L^2 I_d$. From Lemma 15 below, we have

$$\sum_{s=t_0+1}^{t} \|\phi_{k_s}\|_{(V_{t_0+1:s-1} + 2L^2 I_d)^{-1}}^2 = \sum_{s=t_0+1}^{t} \|\phi_{k_s}\|_{(V_{t_0+1:s-1} + L^2 I_d + L^2 I_d)^{-1}}^2$$

$$\leq \sum_{s=t_0+1}^{t} \|\phi_{k_s}\|_{(V_{t_0+1:s} + L^2 I_d)^{-1}}^2$$

$$\leq d \log \left(1 + \frac{2t}{d}\right) .$$

$\qquad \square$

A general statement (extracted from [13] but widely known, see for example [28]) is

**Lemma 15.** *Let $(\omega_t)_{t\geq 1}$ be a sequence in the simplex $\Delta_K$ and $x > 0$. Let $W_t := \sum_{s=1}^{t} \omega_s$ and $V_{W_t} := \sum_{s=1}^{t} \sum_{k=1}^{K} \omega_s^k \phi_k \phi_k^\top$. Then*

$$\sum_{s=1}^{t} \sum_{k=1}^{K} \omega_s^k \|\phi_k\|_{(V_{W_s}+xI_d)^{-1}}^2 \leq d \log\left(1 + \frac{tL^2}{d\eta}\right).$$

*Proof.* Define the function $f(W) = \log\det(V_W + xI_d)$ for any $W \in (\mathbb{R}^+)^K$. It is a concave function since the function $V \mapsto \log\det(V)$ is a concave function over the set of positive definite matrices (see Exercise 21.2 of [28]). Its partial derivative with respect to the coordinate $k$ at $W$ is

$$\nabla_k f(W) = \|\phi_k\|_{(V_W+xI_d)^{-1}}^2 .$$

Hence using the concavity of $f$ we have

$$\sum_{k=1}^{K} \omega_s^k \|\phi_k\|_{(V_{W_s}+xI_d)^{-1}}^2 = (W_s - W_{s-1})^\top \nabla_a f(W_s) \leq f(W_s) - f(W_{s-1}),$$

which implies that

$$\sum_{s=1}^{t} \sum_{k=1}^{K} \omega_s^k \|\phi_k\|_{V_{W_s}+xI_d}^2 \leq f(W_t) - f(W_0) = \log\left(\frac{\det(V_{W_t}+xI_d)}{\det(xI_d)}\right) \leq d\log\left(1 + \frac{tL^2}{dx}\right),$$

where for the last inequality we use the inequality of arithmetic and geometric means in combination with $\text{Tr}(V_{W_t}) \leq tL^2$. □

**Lemma 16.** *Let $C > 0$ be a constant. With probability $1 - \delta$,*

$$\sum_{s=t_0+1}^{t} \sum_{k=1}^{K} \omega_{s-1}^k \min\left\{C, \frac{1}{N_{s-1}^k}\right\} \leq C\sqrt{2t\log\frac{1}{\delta}} + K(C + 1 + \log t)$$

*Proof.* The first term is due to a martingale argument to bound $\sum_s \left(\sum_k \omega_{s-1}^k \min\left\{C, \frac{1}{N_{s-1}^k}\right\} - \min\left\{C, \frac{1}{N_{s-1}^{k_s}}\right\}\right)$. Then

$$\sum_{s=t_0+1}^{t} \min\left\{C, \frac{1}{N_{s-1}^{k_s}}\right\} \leq CK + \sum_{k=1}^{K} \mathbb{I}\left\{N_{t-1}^k > 0\right\} \sum_{j=1}^{N_{t-1}^k} \frac{1}{j} \leq K(C + 1 + \log t).$$

□

### E.4 Martingale concentration

**Lemma 17.** *Let $\mu \in \mathcal{M}$ (with upper bounds $M$ and $\varepsilon$) and $Z_s(\lambda) := (\mu^{k_s} - \lambda^{k_s})^2 - \mathbb{E}_{k\sim\omega_s}[(\mu^k - \lambda^k)^2]$. For any $\delta' \in (0, 1)$,*

$$\mathbb{P}\left\{\exists t \geq 1 : \sup_{\lambda\in\mathcal{M}} \left|\sum_{s=1}^{t} Z_s(\lambda)\right| > r(t, \delta')\right\} \leq \delta',$$

*where*

$$r(t, \delta') := 2M^2\sqrt{\frac{t}{2}\left(\log\frac{4t^2}{\delta'} + d\log\frac{6(M+\varepsilon)Lt}{\sqrt{\Gamma(A)}} + K\log\max\{4\varepsilon t, 1\}\right)} + 2 + 8M,$$

$\Gamma(A) := \max_{\omega\in\Delta_K} \sigma_{\min}\left(\sum_{k=1}^{K} \omega^k \phi_k \phi_k^\top\right)$, *and $\sigma_{\min}(M)$ is the minimal eigenvalue of matrix $M$.*

*Proof.* First note that $\omega_s$ is $\mathcal{F}_{s-1}$-measurable. Thus, for any fixed $\lambda$,

$$\mathbb{E}[Z_s(\lambda)|\mathcal{F}_{s-1}] = \mathbb{E}[(\mu^{k_s} - \lambda^{k_s})^2|\mathcal{F}_{s-1}] - \mathbb{E}_{k\sim\omega_s}[(\mu^k - \lambda^k)^2] = 0,$$

which implies that $\{Z_s\}_{s\geq 1}$ is a martingale difference sequence. Moreover, it is easy to check that $|Z_s(\lambda)| \leq 4M^2$. Unfortunately, we cannot directly use this martingale property to concentrate the desired term since $\lambda$ is adaptively chosen as a function of the whole history up to time $t$. As a solution, we shall use a covering argument on the whole model family $\mathcal{M}$.

Suppose that we have a finite $\xi$-cover $\bar{\mathcal{M}}_\xi$ of $\mathcal{M}$, i.e., for any $\lambda \in \mathcal{M}$, there exists $\bar{\lambda} \in \bar{\mathcal{M}}_\xi$ such that $\|\lambda - \bar{\lambda}\|_\infty \leq \xi$. For such a couple $(\lambda, \bar{\lambda})$, this implies that, for any $s \geq 1, k \in [K]$,

$$\left|(\mu^k - \lambda^k)^2 - (\mu^k - \bar{\lambda}_k)^2\right| = \left|(\bar{\lambda}_k - \lambda^k)^2 + 2(\mu^k - \bar{\lambda}_k)(\bar{\lambda}_k - \lambda^k)\right|$$
$$\leq (\bar{\lambda}_k - \lambda^k)^2 + 2|\mu^k - \bar{\lambda}_k||\bar{\lambda}_k - \lambda^k| \leq \xi^2 + 4M\xi.$$

Moreover, using this bound in the definition of $Z_s(\lambda)$,

$$\left|\sum_{s=1}^t Z_s(\lambda) - \sum_{s=1}^t Z_s(\bar{\lambda})\right| \leq 2t\xi^2 + 8tM\xi.$$

Let $h(t)$ be some function to be specified later. With some abuse of notation w.r.t. the derivation above, we shall instantiate a different $\xi_t$-cover for each time step $t$. Then

$$\mathbb{P}\left\{\exists t \geq 1 : \sup_{\lambda\in\mathcal{M}}\left|\sum_{s=1}^t Z_s(\lambda)\right| > h(t)\right\} = \mathbb{P}\left\{\exists t \geq 1 : \sup_{\lambda\in\mathcal{M}}\inf_{\bar{\lambda}\in\bar{\mathcal{M}}_{\xi_t}}\left|\sum_{s=1}^t Z_s(\lambda) \pm Z_s(\bar{\lambda})\right| > h(t)\right\}$$

$$\overset{(a)}{\leq} \mathbb{P}\left\{\exists t \geq 1 : \sup_{\bar{\lambda}\in\bar{\mathcal{M}}_{\xi_t}}\left|\sum_{s=1}^t Z_s(\bar{\lambda})\right| + \sup_{\lambda\in\mathcal{M}}\inf_{\bar{\lambda}\in\bar{\mathcal{M}}_{\xi_t}}\left|\sum_{s=1}^t Z_s(\lambda) - Z_s(\bar{\lambda})\right| > h(t)\right\}$$

$$\overset{(b)}{\leq} \mathbb{P}\left\{\exists t \geq 1, \bar{\lambda}\in\bar{\mathcal{M}}_{\xi_t} : \left|\sum_{s=1}^t Z_s(\bar{\lambda})\right| > h(t) - 2t{\xi_t}^2 - 8tM\xi_t\right\}$$

$$\overset{(c)}{\leq} \sum_{t=1}^\infty \sum_{\bar{\lambda}\in\bar{\mathcal{M}}_{\xi_t}} \mathbb{P}\left\{\left|\sum_{s=1}^t Z_s(\bar{\lambda})\right| > h(t) - 2t{\xi_t}^2 - 8tM\xi_t\right\},$$

where (a) follows by the triangle inequality, (b) from the property of the cover, and (c) from the union bound and the fact that the cover is finite. Let $\delta'_t \in (0,1)$. If we choose $h(t) := 2M^2\sqrt{\frac{t}{2}\log(2/\delta'_t)} + 2t{\xi_t}^2 + 8tM\xi_t$, using Azuma's inequality, each probability in the sum above is bounded by $\delta'_t$. Hence, choosing $\delta'_t := \frac{\delta'}{2|\bar{\mathcal{M}}_{\xi_t}|t^2}$,

$$\sum_{t=1}^\infty \sum_{\bar{\lambda}\in\bar{\mathcal{M}}_{\xi_t}} \mathbb{P}\left\{\left|\sum_{s=1}^t Z_s(\bar{\lambda})\right| > h(t) - 2t{\xi_t}^2 - 8tM\xi_t\right\} \leq \sum_{t=1}^\infty \frac{\delta'}{2t^2} \leq \delta',$$

where the last inequality can be verified easily. Therefore, putting everything together, we proved that

$$\mathbb{P}\left\{\exists t \geq 1 : \sup_{\lambda\in\mathcal{M}}\left|\sum_{s=1}^t Z_s(\lambda)\right| > 2M^2\sqrt{\frac{t}{2}\log\frac{4|\bar{\mathcal{M}}_{\xi_t}|t^2}{\delta'}} + 2t{\xi_t}^2 + 8tM\xi_t\right\} \leq \delta'.$$

It only remains to build the cover, compute its size, and specify the value of $\xi_t$. Recall that each model $\lambda \in \mathcal{M}$ can be written as $\lambda = A\theta' + \eta'$, where $\|\eta'\|_\infty \leq \varepsilon$ and $\|\lambda\|_\infty \leq M$. Using Lemma 28 below, we have that $\|\theta'\|_2 \leq \bar{B} := \frac{M+\varepsilon}{\sqrt{\Gamma(A)}}$. Then, we can build two separate covers for the linear and deviation parts. Specifically, we build a $\xi_t/(2L)$-cover $\bar{\mathcal{M}}_t^{\mathrm{lin}}$ in $\ell_2$-norm for the linear part and a $\xi_t/2$-cover $\bar{\mathcal{M}}_t^{\mathrm{dev}}$ in $\ell_\infty$-norm for the deviation part. Then, we take the full cover as the (finite) set $\bar{\mathcal{M}}_t := \{\bar{\lambda} = A\bar{\theta} + \bar{\eta} : \bar{\theta} \in \bar{\mathcal{M}}_t^{\mathrm{lin}}, \bar{\eta} \in \bar{\mathcal{M}}_t^{\mathrm{dev}}\}$. With this choice, we have that, for any $\lambda = A\theta' + \eta'$, there exits $\bar{\lambda} \in \bar{\mathcal{M}}_t$ such that

$$\|\lambda - \bar{\lambda}\|_\infty = \|A\theta' + \eta' - A\bar{\theta} - \bar{\eta}\|_\infty \leq L\|\theta' - \bar{\theta}\|_2 + \|\eta' - \bar{\eta}\|_\infty \leq \xi_t.$$

Let us compute the size of the cover $\bar{\mathcal{M}}_t$. It is easy to see that this is $|\bar{\mathcal{M}}_t| = |\bar{\mathcal{M}}_t^{\text{lin}}||\bar{\mathcal{M}}_t^{\text{dev}}|$. For the linear one, it is known that the $\xi_t/(2L)$-covering number (in $\ell_2$-norm) of a ball in $\mathbb{R}^d$ with radius $\bar{B}$ is at most $(6L\bar{B}/\xi_t)^d$. For the deviation, we can have a $\xi_t/2$ cover in $\ell_\infty$-norm with at most $\max\{(4\varepsilon/\xi_t)^K, 1\}$ points, where the maximum is to deal with too small values of $\varepsilon$ (e.g., $\varepsilon = 0$). Then, the final cover has size at most $|\bar{\mathcal{M}}_t| \leq (6\bar{B}L/\xi_t)^d \max\{(4\varepsilon/\xi_t)^K, 1\}$. Setting $\xi_t = 1/t$, we get the desired bound. $\qquad\square$

# F $\delta$-correctness and sample complexity analysis

## F.1 Correctness

We prove Lemma 3 in the main paper, restated below.

**Lemma.** *Let $W_{-1}$ be the negative branch of the Lambert $W$ function and let $\overline{W}(x) = -W_{-1}(-e^{-x}) \approx x + \log x$. For $\delta \in (0, 1)$, define*

$$\beta_{t,\delta}^{\text{uns}} := 2K\overline{W}\left(\frac{1}{2K}\log\frac{2e}{\delta} + \frac{1}{2}\log(8eK\log t)\right), \tag{7}$$

$$\beta_{t,\delta}^{\text{lin}} := \frac{1}{2}\left(4\sqrt{t}\varepsilon + \sqrt{2}\sqrt{1 + \log\frac{1}{\delta} + \left(1 + \frac{1}{\log(1/\delta)}\right)\frac{d}{2}\log\left(1 + \frac{t}{2d}\log\frac{1}{\delta}\right)}\right)^2. \tag{8}$$

*Then, for the choice $\beta_{t,\delta} := \min\{\beta_{t,\delta}^{\text{uns}}, \beta_{t,\delta}^{\text{lin}}\}$, MISLID is $\delta$-correct.*

*Proof.* $\delta$-correctness is composed of two properties: stopping in a finite time with probability one and verifying, for all instances $\mu \in \mathcal{M}$, $\mathbb{P}(\hat{S}_m \not\subseteq S^\star(\mu)) \leq \delta$. The fact that the stopping time is finite almost surely is a consequence of the sample complexity bound (see further down in this section). We now prove the bound on the probability of error in identification.

We first relate the event that the algorithm does not return a correct answer to a large deviation, by writing that for the algorithm to make a mistake, there must be a time at which the stopping condition is met and $\tilde{\mu}_t$ is in the alternative to $\mu$:

$$\mathbb{P}(\hat{S}_m \not\subseteq S^\star(\mu)) \leq \mathbb{P}\left(\exists t \in \mathbb{N}, \inf_{\lambda \in \Lambda_m(\tilde{\mu}_t)} \|\tilde{\mu}_t - \lambda\|_{D_{N_t}}^2 > 2\beta_{t,\delta} \wedge \tilde{\mu}_t \in \Lambda_m(\mu)\right).$$

If the two conditions of the right-hand side happen, then $\mu \in \Lambda_m(\tilde{\mu}_t)$ and we get

$$\mathbb{P}(\hat{S}_m \not\subseteq S^\star(\mu)) \leq \mathbb{P}\left(\exists t \in \mathbb{N}, \|\tilde{\mu}_t - \mu\|_{D_{N_t}}^2 > 2\beta_{t,\delta}\right).$$

It then suffices to prove that we have both

$$\mathbb{P}\left(\exists t \in \mathbb{N}, \frac{1}{2}\|\tilde{\mu}_t - \mu\|_{D_{N_t}}^2 > \beta_{t,\delta}^{\text{lin}}\right) \leq \delta/2, \tag{9}$$

and $\quad \mathbb{P}\left(\exists t \in \mathbb{N}, \frac{1}{2}\|\tilde{\mu}_t - \mu\|_{D_{N_t}}^2 > \beta_{t,\delta}^{\text{uns}}\right) \leq \delta/2. \tag{10}$

The result for (10) is Lemma 11 (and the remark below that lemma stating that it applies to $\tilde{\mu}_t$). We now prove the concentration inequality using the linear term (9).

let $\tilde{\theta}_{t,\varepsilon}$ and $\tilde{\eta}_{t,\varepsilon}$ be parameters for $\tilde{\mu}_t$ with $\|\tilde{\eta}_{t,\varepsilon}\| \leq \varepsilon$, which exist since $\tilde{\mu}_t \in \mathcal{M}$. On the other hand, let $\tilde{\theta}_t$ and $\tilde{\eta}_t$ be the orthogonal parameters of $\tilde{\mu}_t$ with respect to $N_t$.

$$\begin{aligned}
\|\tilde{\mu} - \mu\|_{D_{N_t}} &= \|A(\tilde{\theta}_{t,\varepsilon} - \theta) + \tilde{\eta}_{t,\varepsilon} - \eta\|_{D_{N_t}} \\
&\leq \|A(\tilde{\theta}_{t,\varepsilon} - \theta)\|_{D_{N_t}} + \|\tilde{\eta}_{t,\varepsilon} - \eta\|_{D_{N_t}} \\
&= \|\tilde{\theta}_{t,\varepsilon} - \theta\|_{V_t} + \|\tilde{\eta}_{t,\varepsilon} - \eta\|_{D_{N_t}} \\
&\leq \|\tilde{\theta}_{t,\varepsilon} - \tilde{\theta}_t\|_{V_t} + \|\tilde{\theta}_t - \theta_t\|_{V_t} + \|\theta_t - \theta\|_{V_t} + \|\tilde{\eta}_{t,\varepsilon} - \eta\|_{D_{N_t}}.
\end{aligned}$$

Lemma 6 bounds the first and third terms by $\sqrt{t}\varepsilon$. The last term is bounded by $\sqrt{t}\|\tilde{\eta}_{t,\varepsilon}-\eta\|_\infty \le 2\sqrt{t}\varepsilon$ since both vectors have $\ell_\infty$ norm bounded by $\varepsilon$.

Finally

$$\mathbb{P}\left(\exists t \in \mathbb{N},\ \frac{1}{2}\left\|\tilde{\mu}_t - \mu\right\|_{D_{N_t}}^2 > \beta_{t,\delta}^{\mathrm{lin}}\right) \le \mathbb{P}\left(\exists t \in \mathbb{N},\ \frac{1}{2}\left\|\widehat{\theta}_t - \theta_t\right\|_{V_t}^2 > \frac{1}{2}\left(\sqrt{2\beta_{t,\delta}^{\mathrm{lin}}} - 4\sqrt{t}\varepsilon\right)^2\right) \le \delta/2$$

by Corollary 2.

$\square$

## F.2   Restriction to a good event

**Assumption.**   We start by pulling arms deterministically until $t_0$, such that $V_{t_0} \ge 2L^2 I_d$. See paragraph "Initialization phase" in Subsection 4.1 in the main paper.

**Definition of the good event.**   For $t \ge t_0$ and $k \in [K]$, define

$$\alpha_t^{\mathrm{lin}} := \log(5t^2) + d\log\left(1 + \frac{t}{2d}\right)\ ,\quad \alpha_t^{\mathrm{uns}} := 2K\overline{W}\left(\frac{1}{2K}\log(2e5t^3) + \frac{1}{2}\log(8eK\log t)\right)\ .$$

Consider the following events. Each of these holds with probability at least $1 - \frac{1}{5t^2}$ by the indicated concentration result.

1. Concentration of the projected linear part (Corollary 1)

$$\mathcal{E}_t^1 := \left\{\forall s \ge t_0:\ \frac{1}{2}\left\|\tilde{\theta}_s - \theta_s\right\|_{V_s}^2 \le \alpha_t^{\mathrm{lin}}\right\}\ ,$$

2. Unstructured concentration of the projected estimator (Lemma 11)

$$\mathcal{E}_t^2 := \left\{\forall s \le t:\ \frac{1}{2}\left\|\tilde{\mu}_s - \mu\right\|_{D_{N_s}}^2 \le \alpha_t^{\mathrm{uns}}\right\}\ ,$$

3. Martingale concentration for sampling (Lemma 17)

$$\mathcal{E}_t^3 := \left\{\sup_{\lambda \in \mathcal{M}}\left|\sum_{s=1}^t Z_s(\lambda)\right| \le r(t)\right\}\ ,$$

where $r(t)$ is obtained by setting $\delta' = \frac{1}{5t^2}$ in $r(t,\delta')$ in Lemma 17, which yields

$$r(t) = 2M^2\sqrt{\frac{t}{2}\left(\log(4 \times 5t^4) + d\log\frac{6(M+\varepsilon)Lt}{\sqrt{\Gamma(A)}} + K\log\max\{4\varepsilon t, 1\}\right)} + 2 + 8M.$$

4. Elliptical potential with sampling (Lemma 12)

$$\mathcal{E}_t^4 := \left\{\sum_{s=t_0+1}^t \sum_{k=1}^K \omega_s^k \|\phi_k\|_{V_{s-1}^{-1}}^2 \le \sqrt{2t\log(5t^2)} + d\log\left(1 + \frac{t}{d}\right)\right\}\ ,$$

5. Elliptical potential with sampling for the unstructured bound (Lemma 16)

$$\mathcal{E}_t^5 := \left\{\sum_{s=t_0+1}^t \sum_{k=1}^K \omega_s^k \min\left\{4M^2, \frac{2\alpha_t^{\mathrm{uns}}}{N_{s-1}^k}\right\} \le 4M^2\sqrt{2t\log(5t^2)} + 4M^2K + 2K\alpha_t^{\mathrm{uns}}\log(et)\right\}\ .$$

Then, we define the "good" event $\mathcal{E}_t := \bigcap_{i=1}^5 \mathcal{E}_t^i$.

**Lemma 18.** *For all $t \ge 1$, $\mathbb{P}(\mathcal{E}_t^c) \le 1/t^2$ .*

*Proof.* Apply an union bound by noting that each event $\mathcal{E}_t^i$ fails with probability at most $1/(5t^2)$. $\square$

**Lemma 19.** *Let $T_0(\delta) \in \mathbb{N}$ be such that for $t \geq T_0(\delta)$, $\mathcal{E}_t \subseteq \{\tau_\delta \leq t\}$. Then $\mathbb{E}[\tau_\delta] \leq T_0(\delta) + 2$.*

*Proof.* Successively using the definition of $T_0(\delta)$ and Lemma 18:

$$\mathbb{E}[\tau_\delta] = \sum_{t=0}^{+\infty} \mathbb{P}(\tau_\delta > t) \leq T_0(\delta) + \sum_{t=T_0(\delta)}^{+\infty} \mathbb{P}(\mathcal{E}_t^c) \leq T_0(\delta) + \sum_{t=1}^{+\infty} \frac{1}{t^2} \leq T_0(\delta) + 2.$$

$\square$

**Consequences of the good event.**

**Lemma 20.** *For $t \geq t_0$ and $k \in [K]$, define*

$$c_t^k := \min\left\{ 8(LK+1)^2 \varepsilon^2 + 4\alpha_{t^2}^{\mathrm{lin}} \|\phi_k\|_{V_t^{-1}}^2, \frac{2\alpha_{t^2}^{\mathrm{uns}}}{N_t^k}, 4M^2 \right\},$$

*where we use the convention that $2\alpha^{\mathrm{uns}}/N_t^k = +\infty$ if $N_t^k = 0$. Then under $\mathcal{E}_t$, for all $s \in \{\max\{t_0, \sqrt{t}\}, \ldots, t\}$ and $k \in [K]$, $(\tilde{\mu}_s^k - \mu^k)^2 \leq c_s^k$.*

*Proof.* We know that $\frac{1}{2} \left\| \tilde{\theta}_s - \theta_s \right\|_{V_t}^2 \leq \alpha_t^{\mathrm{lin}}$ holds for all $s \geq t_0$ by definition of $\mathcal{E}_t^1$. For $s \geq \max\{t_0, \sqrt{t}\}$ we also have $\alpha_{s^2}^{\mathrm{lin}} \geq \alpha_t^{\mathrm{lin}}$, hence $\frac{1}{2} \left\| \tilde{\theta}_s - \theta_s \right\|_{V_t}^2 \leq \alpha_{s^2}^{\mathrm{lin}}$. Using first $(a+b)^2 \leq 2a^2 + 2b^2$ then the Cauchy-Schwarz inequality on $(V_s^{-1/2} \phi_k)^\top (V_s^{1/2}(\tilde{\theta}_s - \theta_s))$ and Lemma 5,

$$(\tilde{\mu}_s^k - \mu^k)^2 \leq 2(\phi_k^\top(\tilde{\theta}_s - \theta_s))^2 + 2(\tilde{\eta}_s^k - \eta_s^k)^2 \leq 2 \|\phi_k\|_{V_s^{-1}}^2 \left\| \tilde{\theta}_s - \theta_s \right\|_{V_s}^2 + 8(LK+1)^2 \varepsilon^2$$

$$\leq 8(LK+1)^2 \varepsilon^2 + 4\alpha_{s^2}^{\mathrm{lin}} \|\phi_k\|_{V_s^{-1}}^2.$$

Moreover by definition of $\mathcal{E}_t^2$, for all $s \leq t$, $\frac{1}{2} \|\tilde{\mu}_s - \mu\|_{D_{N_s}}^2 \leq \alpha_t^{\mathrm{uns}}$. For $s \geq \max\{t_0, \sqrt{t}\}$ we have $\alpha_{s^2}^{\mathrm{uns}} \geq \alpha_t^{\mathrm{uns}}$, hence $\frac{1}{2} \|\tilde{\mu}_s - \mu\|_{D_{N_s}}^2 \leq \alpha_{s^2}^{\mathrm{uns}}$. Therefore,

$$(\tilde{\mu}_s^k - \mu^k)^2 = (e_k^\top(\tilde{\mu}_s - \mu))^2 \leq \|e_k\|_{D_{N_s}^{-1}}^2 \|\mu - \tilde{\mu}_s\|_{D_{N_s}}^2 \leq \frac{2\alpha_{s^2}^{\mathrm{uns}}}{N_s^k}.$$

Finally, $(\tilde{\mu}_s^k - \mu^k)^2 \leq \|\tilde{\mu}_s - \mu\|_\infty^2 \leq 4M^2$. $\square$

**Lemma 21.** *For all $t \geq 1$, under the good event $\mathcal{E}_t$,*

$$\forall s \in \{t_0, t_0 + 1, \ldots, t\} : \|\tilde{\mu}_s - \mu\|_{D_{N_s}}^2 \leq f(t) := 2\min\{\alpha_t^{\mathrm{uns}}, \alpha_t^{\mathrm{lin}} + 2t(LK+1)^2 \varepsilon^2\}.$$

*Proof.* That for all $s \leq t$, $\|\tilde{\mu}_s - \mu\|_{D_{N_s}}^2 \leq 2\alpha_t^{\mathrm{uns}}$ directly follows from the definition of $\mathcal{E}_t^2$. To see the second inequality, we first decompose the norm on the lefthand-side into its linear and deviation components

$$\|\tilde{\mu}_s - \mu\|_{D_{N_s}}^2 = \left\| \tilde{\theta}_s - \theta_s \right\|_{V_s}^2 + \left\| R_{N_s} D_{N_s}^{1/2} \tilde{\eta}_s - R_{N_s} D_{N_s}^{1/2} \eta_s \right\|^2.$$

The deviation part can be bounded by $4t(LK+1)^2 \varepsilon^2$ for all $s \leq t$ using Lemma 5. The linear part can be bounded by $2\alpha_t^{\mathrm{lin}}$ for all $s \geq t_0$ by the definition of $\mathcal{E}_t^1$. $\square$

## F.3 Analysis under a good event

Fix any time step $t \geq t_0$. Suppose that the good event $\mathcal{E}_t$ of Section F.2 holds and the algorithm does not stop at time $t$. We proceed in different steps.

**Stopping rule analysis.**

**Theorem 4.** *If the algorithm does not stop at time $t$ then under $\mathcal{E}_t$, using stopping threshold $\beta_{t,\delta}$ as defined in Lemma 3 in the main paper,*

$$2\beta_{t,\delta} \geq \inf_{\lambda \in \Lambda_m(\mu)} \|\mu - \lambda\|^2_{D_{W_t}} - h_\delta(t) - r(t) \, .$$

*where*

- $h_\delta(t) = \sqrt{8\beta_{t,\delta} f(t)} + f(t)$, *with* $f(t)$ *a bound on* $\|\mu - \tilde{\mu}_t\|^2_{D_{N_t}}$ *(see Lemma 21)* ,

- $r(t) = 2M^2 \sqrt{\frac{t}{2}\left(\log(4 \times 5t^4) + d\log\frac{6(M+\varepsilon)Lt}{\sqrt{\Gamma(A)}} + K\log\max\{4\varepsilon t, 1\}\right)} + 2 + 8M$ ,

*and $W_t := \sum_{s \leq 1}^t \omega_s$ is the sum over time of the weight vectors played by the learner.*

The proof of this theorem is detailed in Steps 1 to 3 below.

**Step 1. From $\Lambda_m(\tilde{\mu}_t)$ to $\Lambda_m(\mu)$.**

**Lemma 22.** *For all $\mu, \mu' \in \mathcal{M}$, for any non-negative function $f : \mathbb{R}^K \times \mathbb{R}^K \to \mathbb{R}$ with $f(x,x) = 0$,*

$$\inf_{\lambda \in \Lambda_m(\mu)} f(\mu, \lambda) \geq \inf_{\lambda \in \Lambda_m(\mu')} f(\mu, \lambda) \, .$$

*Proof.* Either $\Lambda_m(\mu) = \Lambda_m(\mu')$ and the two expressions are equal, or $\Lambda_m(\mu) \neq \Lambda_m(\mu')$. In the second case, $\mu \in \Lambda_m(\mu')$. The right-hand side is then equal to zero, which is lower than the left-hand side since $f$ is non-negative. $\square$

Since the algorithm does not stop at time $t$, from the stopping rule

$$2\beta_{t,\delta} \geq \inf_{\lambda \in \Lambda_m(\tilde{\mu}_t)} \|\tilde{\mu}_t - \lambda\|^2_{D_{N_t}} \, ,$$

where $\Lambda_m(\tilde{\mu}_t)$ is the set of alternative models to $\tilde{\mu}_t$. We change the alternative set over which the minimization is performed using Lemma 22:

$$2\beta_{t,\delta} \geq \inf_{\lambda \in \Lambda_m(\tilde{\mu}_t)} \|\tilde{\mu}_t - \lambda\|^2_{D_{N_t}} \geq \inf_{\lambda \in \Lambda_m(\mu)} \|\tilde{\mu}_t - \lambda\|^2_{D_{N_t}} \, . \tag{11}$$

**Step 2. From $\tilde{\mu}_t$ to $\mu$.** The next step is to replace the estimated mean $\tilde{\mu}_t$ in the norm with the true mean $\mu$. For all $\lambda \in \mathcal{M}$, using the triangle inequality,

$$\|\tilde{\mu}_t - \lambda\|_{D_{N_t}} \geq \|\mu - \lambda\|_{D_{N_t}} - \|\tilde{\mu}_t - \mu\|_{D_{N_t}} \geq \|\mu - \lambda\|_{D_{N_t}} - \sqrt{f(t)} \, ,$$

where the last inequality uses Lemma 21 to concentrate $\|\tilde{\mu}_t - \mu\|^2_{D_{N_t}}$. Using this for the specific choice of $\lambda_t \in \arg\min_{\lambda \in \Lambda_m(\mu)} \|\tilde{\mu}_t - \lambda\|^2_{D_{N_t}}$ in combination with (11), we obtain

$$\left(\sqrt{2\beta_{t,\delta}} + \sqrt{f(t)}\right)^2 \geq \|\mu - \lambda_t\|^2_{D_{N_t}} \quad \Rightarrow \quad 2\beta_{t,\delta} \geq \inf_{\lambda \in \Lambda_m(\mu)} \|\mu - \lambda\|^2_{D_{N_t}} - h_\delta(t) \, , \tag{12}$$

where $h_\delta(t) := \sqrt{8\beta_{t,\delta} f(t)} + f(t)$ is a sub-linear function of both $t$ and $\log(1/\delta)$.

**Step 3. From $N_t$ to $W_t$.** We now show that it is possible to replace $N_t$ with $W_t := \sum_{s=1}^t \omega_s$ in the norm at the price of subtracting another low-order term. Let $Z_s(\lambda) := (\mu^{k_s} - \lambda^{k_s})^2 - \mathbb{E}_{k \sim \omega_s}[(\mu^k - \lambda^k)^2]$. Note that $\|\mu - \lambda\|^2_{D_{N_t}} = \sum_{s=1}^t (\mu^{k_s} - \lambda^{k_s})^2$ and $\|\mu - \lambda\|^2_{D_{W_t}} = \sum_{s=1}^t \|\mu - \lambda\|^2_{D_{\omega_s}} = \sum_{s=1}^t \mathbb{E}_{k \sim \omega_s}[(\mu^k - \lambda^k)^2]$. Therefore, from (12),

$$2\beta_{t,\delta} \geq \inf_{\lambda \in \Lambda_m(\mu)} \left(\|\mu - \lambda\|^2_{D_{N_t}} - \|\mu - \lambda\|^2_{D_{W_t}} + \|\mu - \lambda\|^2_{D_{W_t}}\right) - h_\delta(t)$$

$$= \inf_{\lambda \in \Lambda_m(\mu)} \left(\|\mu - \lambda\|^2_{D_{W_t}} + \sum_{s=1}^t Z_s(\lambda)\right) - h_\delta(t)$$

$$\geq \inf_{\lambda \in \Lambda_m(\mu)} \|\mu - \lambda\|^2_{D_{W_t}} - \sup_{\lambda \in \mathcal{M}} \left|\sum_{s=1}^t Z_s(\lambda)\right| - h_\delta(t) \, .$$

Using the good event $\mathcal{E}_t^3$, we can finally write

$$2\beta_{t,\delta} \geq \inf_{\lambda \in \Lambda_m(\mu)} \|\mu - \lambda\|_{D_{W_t}}^2 - h_\delta(t) - r(t) \,, \tag{13}$$

which ends proving Theorem 4.

**Sampling rule analysis.** Let $H_\mu = \sup_{\omega \in \Delta} \inf_{\lambda \in \Lambda_m(\mu)} \frac{1}{2} \|\mu - \lambda\|_{D_\omega}^2$ (the inverse complexity at $\mu$). In the first part of the sampling rule analysis, we introduce the optimistic estimates $g_t(\omega)$ mentioned in Algorithm 1 in the main paper, which will be used by the learner for $\omega_t$.

**Theorem 5.** *Let $(\tilde{\mu}_s)_{s \leq t} \in \mathcal{M}^{[t]}$ be estimates such that under $\mathcal{E}_t$, we have a bound $c_s^k$ on $(\tilde{\mu}_s^k - \mu^k)^2$ for all $k \in [K]$ and $s \in [t]$. Then define the* optimistic estimate

$$g_s(\omega) := \sum_{k=1}^K \omega^k \left( \left| \tilde{\mu}_{s-1}^k - \lambda_s^k \right| + \sqrt{c_{s-1}^k} \right)^2 \quad where \quad \lambda_s := \operatorname*{arg\,min}_{\lambda \in \Lambda_m(\tilde{\mu}_{s-1})} \|\tilde{\mu}_{s-1} - \lambda\|_{D_{\omega_s}}^2 \,.$$

*Under $\mathcal{E}_t$,*

$$\inf_{\lambda \in \Lambda_m(\mu)} \|\mu - \lambda\|_{D_{W_t}}^2 \geq \sum_{s=t_0+1}^t g_s(\omega_s) - 4C_t - 4\sqrt{2tC_tH_\mu} \,,$$

*with $C_t := \sum_{s=t_0+1}^t \sum_{k=1}^K \omega_s^k c_{s-1}^k$.*

The proof of this theorem is detailed in the Steps 4 to 7 below. Once this result is established, we will use the regret property of the learner to exhibit the final bound (Steps 8 to 10).

**Step 4. From $\Lambda_m(\mu)$ back to $\Lambda_m(\tilde{\mu}_{s-1})$ for $s \in [t]$.** We now start moving from $\inf_{\lambda \in \Lambda_m(\mu)} \|\mu - \lambda\|_{D_{W_t}}^2$ to the actual gain fed into the online learner at time $t$. We first need to go back to the estimated set of alternative models at each time $s = 0, \ldots, t-1$. We have

$$\inf_{\lambda \in \Lambda_m(\mu)} \|\mu - \lambda\|_{D_{W_t}}^2 = \inf_{\lambda \in \Lambda_m(\mu)} \sum_{s=1}^t \|\mu - \lambda\|_{D_{\omega_s}}^2 \geq \sum_{s=1}^t \inf_{\lambda \in \Lambda_m(\mu)} \|\mu - \lambda\|_{D_{\omega_s}}^2 \tag{14}$$

$$\geq \sum_{s=1}^t \inf_{\lambda \in \Lambda_m(\tilde{\mu}_{s-1})} \|\mu - \lambda\|_{D_{\omega_s}}^2 \,, \tag{15}$$

where the first inequality follows by the concavity of the infimum, and the second one is an application of Lemma 22.

**Step 5. Drop the first rounds.** The first $t_0$ rounds are dedicated to making sure that $V_t$ is sufficiently large (for the partial order on positive definite matrices). Also, our upper bounds on the deviation of $\tilde{\mu}_t^k$ from $\mu^k$ are valid from $\max\{t_0, \sqrt{t}\}$. We define $t_0'(t) = \max\{t_0, \sqrt{t}\}$. We drop the corresponding nonnegative terms from the sum to keep only the rounds for which $t$ is large enough:

$$\sum_{s=1}^t \inf_{\lambda \in \Lambda_m(\tilde{\mu}_{s-1})} \|\mu - \lambda\|_{D_{\omega_s}}^2 \geq \sum_{s=t_0'(t)+1}^t \inf_{\lambda \in \Lambda_m(\tilde{\mu}_{s-1})} \|\mu - \lambda\|_{D_{\omega_s}}^2 \,.$$

**Step 6. From $\mu$ back to $\tilde{\mu}_{s-1}$ for $s \in [t]$.** We can now use the concentration of $\tilde{\mu}_{s-1}$ to replace $\mu$ in all terms $\|\mu - \lambda\|_{D_{\omega_s}}^2$ for $s \in [t]$. Let $\lambda_s^\mu := \operatorname*{arg\,min}_{\lambda \in \Lambda_m(\tilde{\mu}_{s-1})} \|\mu - \lambda\|_{D_{\omega_s}}^2$. Using first the triangle inequality, then the inequality $\|a - b\| \geq \|a\| - \|b\|$ for an $\ell_2$ norm in dimension $t - t_0'(t)$,

$$\sqrt{\sum_{s=t_0'(t)+1}^t \|\mu - \lambda_s^\mu\|_{D_{\omega_s}}^2} \geq \sqrt{\sum_{s=t_0'(t)+1}^t \left( \|\tilde{\mu}_{s-1} - \lambda_s^\mu\|_{D_{\omega_s}} - \|\mu - \tilde{\mu}_{s-1}\|_{D_{\omega_s}} \right)^2}$$

$$\geq \sqrt{\sum_{s=t_0'(t)+1}^t \|\tilde{\mu}_{s-1} - \lambda_s^\mu\|_{D_{\omega_s}}^2} - \sqrt{\sum_{s=t_0'(t)+1}^t \|\mu - \tilde{\mu}_{s-1}\|_{D_{\omega_s}}^2} \,.$$

We now remark that $\sum_{s=t'_0(t)+1}^{t} \|\tilde{\mu}_{s-1} - \mu\|^2_{D_{w_s}} \leq C_t$ and get, by the definition of $\lambda^\mu_s$,

$$\sqrt{\sum_{s=t'_0(t)+1}^{t} \|\mu - \lambda^\mu_s\|^2_{D_{\omega_s}}} + \sqrt{C_t} \geq \sqrt{\sum_{s=t'_0(t)+1}^{t} \|\tilde{\mu}_{s-1} - \lambda^\mu_s\|^2_{D_{w_s}}} \tag{16}$$

$$\geq \sqrt{\sum_{s=t'_0(t)+1}^{t} \inf_{\lambda \in \Lambda_m(\tilde{\mu}_{s-1})} \|\tilde{\mu}_{s-1} - \lambda\|^2_{D_{w_s}}}. \tag{17}$$

**Step** 7. **Optimistic gains.**  We now replace the term on the right-hand side in (17) by the optimistic gains fed into the online learner. At time $s$, we define optimistic estimates

$$g_s(\omega) := \sum_{k=1}^{K} \omega^k \left( \left| \tilde{\mu}^k_{s-1} - \lambda^k_s \right| + \sqrt{c^k_{s-1}} \right)^2 \quad \text{where } \lambda_s := \arg\min_{\lambda \in \Lambda_m(\tilde{\mu}_{s-1})} \|\tilde{\mu}_{s-1} - \lambda\|^2_{D_{\omega_s}}.$$

**Lemma 23.** *For all $\omega \in \Delta_K$ and $s \geq t'_0(t)$, $g_s(\omega) \geq \inf_{\lambda \in \Lambda_m(\mu)} \|\mu - \lambda\|^2_{D_\omega}$.*

*Proof.* For all $k \in [K]$, $s > t'_0(t)$, and $\lambda \in \mathbb{R}^K$, using Lemma 20 (to write $(\mu^k - \tilde{\mu}^k_{s-1})^2 \leq c^k_{s-1}$):

$$\left( \mu^k - \lambda^k \right)^2 = \left( \tilde{\mu}^k_{s-1} - \lambda^k + \mu^k - \tilde{\mu}^k_{s-1} \right)^2 \leq \left( \left| \tilde{\mu}^k_{s-1} - \lambda^k \right| + \left| \mu^k - \tilde{\mu}^k_{s-1} \right| \right)^2$$

$$\leq \left( \left| \tilde{\mu}^k_{s-1} - \lambda^k \right| + \sqrt{c^k_{s-1}} \right)^2.$$

Then, for any $\omega \in \Delta_K$, by noticing that function $f : \lambda \mapsto \sum_{k}^{K} \omega^k \left( \left| \tilde{\mu}^k_{s-1} - \lambda^k \right|^2 + 2 \left| \tilde{\mu}^k_{s-1} - \lambda^k \right| \sqrt{c^k_{s-1}} \right)$ is nonnegative and that $f(\tilde{\mu}^k_{s-1}) = 0$:

$$g_s(\omega) := \sum_{k=1}^{K} \omega^k \left( \left| \tilde{\mu}^k_{s-1} - \lambda^k_s \right| + \sqrt{c^k_{s-1}} \right)^2$$

$$\geq \inf_{\lambda \in \Lambda_m(\tilde{\mu}_{s-1})} \sum_{k=1}^{K} \omega^k \left( \left| \tilde{\mu}^k_{s-1} - \lambda^k \right| + \sqrt{c^k_{s-1}} \right)^2$$

$$= \sum_{k=1}^{K} \omega^k c^k_{s-1} + \inf_{\lambda \in \Lambda_m(\tilde{\mu}_{s-1})} \sum_{k=1}^{K} \omega^k \left( \left| \tilde{\mu}^k_{s-1} - \lambda^k \right|^2 + 2 \left| \tilde{\mu}^k_{s-1} - \lambda^k \right| \sqrt{c^k_{s-1}} \right)$$

$$\geq \sum_{k=1}^{K} \omega^k c^k_{s-1} + \inf_{\lambda \in \Lambda_m(\mu)} \sum_{k=1}^{K} \omega^k \left( \left| \tilde{\mu}^k_{s-1} - \lambda^k \right|^2 + 2 \left| \tilde{\mu}^k_{s-1} - \lambda^k \right| \sqrt{c^k_{s-1}} \right)$$

(due to Lemma 22)

$$= \inf_{\lambda \in \Lambda_m(\mu)} \sum_{k=1}^{K} \omega^k \left( \left| \tilde{\mu}^k_{s-1} - \lambda^k \right| + \sqrt{c^k_{s-1}} \right)^2$$

$$\geq \inf_{\lambda \in \Lambda_m(\mu)} \sum_{k=1}^{K} \omega^k \left( \mu^k - \lambda^k \right)^2 \quad \text{(using the previously derived coordinate-wise majoration)}$$

$$= \inf_{\lambda \in \Lambda_m(\mu)} \|\mu - \lambda\|^2_{D_\omega}.$$

$\square$

We now prove an upper bound on $g_w(\omega)$, which will be useful later.

**Lemma 24.** *For all $s \geq t_0$ and all $\omega \in \Delta_K$, $g_s(\omega) \leq 36M^2$*

*Proof.* Using the definition of $c_t^k, k \in [K], t \geq 0$ in Lemma 20, and $\mu, \lambda_s \in \mathcal{M}$: $g_s(\omega) = \sum_{k=1}^{K} \omega^k \left( \left| \tilde{\mu}_{s-1}^k - \lambda_s^k \right| + \sqrt{c_{s-1}^k} \right)^2 \leq \sum_{k=1}^{K} \omega^k \left( \left| \mu^k - \lambda_s^k \right| + 2\sqrt{c_{s-1}^k} \right)^2 \leq \sum_{k=1}^{K} \omega^k \left( 6M \right)^2 = 36M^2.$ $\qquad\square$

We have proved that the estimates are indeed optimistic in the sense that they are an upper-bound to the value of interest, as mentioned in paragraph "Optimistic gains" in Subsection 4.1 in the main paper. We now bound by how much they overestimate the empirical value.

**Lemma 25.**

$$\sqrt{\sum_{s=t_0'(t)+1}^{t} \inf_{\lambda \in \Lambda_m(\tilde{\mu}_{s-1})} \|\tilde{\mu}_{s-1} - \lambda\|_{D_{\omega_s}}^2} \geq \sqrt{\sum_{s=t_0'(t)+1}^{t} g_s(\omega_s)} - \sqrt{C_t} \,. \tag{18}$$

*Proof.* We start by a bound for a single $s \in \mathbb{N}$. Using the triangle inequality for an $\ell_2$ norm,

$$\sqrt{g_s(\omega_s)} = \sqrt{\sum_{k=1}^{K} \omega_s^k \left( \left| \tilde{\mu}_{s-1}^k - \lambda_s^k \right| + \sqrt{c_{s-1}^k} \right)^2}$$

$$\leq \sqrt{\sum_{k=1}^{K} \omega_s^k \left( \tilde{\mu}_{s-1}^k - \lambda_s^k \right)^2} + \sqrt{\sum_{k=1}^{K} \omega_s^k c_{s-1}^k} \,.$$

Reordering this inequality, we get

$$\inf_{\lambda \in \Lambda_m(\tilde{\mu}_{s-1})} \|\tilde{\mu}_{s-1} - \lambda\|_{D_{\omega_s}}^2 \geq \left( \sqrt{g_s(\omega_s)} - \sqrt{\sum_{k=1}^{K} \omega_s^k c_{s-1}^k} \right)^2 .$$

Then, summing over $s \in [t_0'(t)+1, t]$ and using $\|a - b\| \geq \|a\| - \|b\|$,

$$\sqrt{\sum_{s=t_0'(t)+1}^{t} \inf_{\lambda \in \Lambda_m(\tilde{\mu}_{s-1})} \|\tilde{\mu}_{s-1} - \lambda\|_{D_{\omega_s}}^2} \geq \sqrt{\sum_{s=t_0'(t)+1}^{t} \left( \sqrt{g_s(\omega_s)} - \sqrt{\sum_{k=1}^{K} \omega_s^k c_{s-1}^k} \right)^2}$$

$$\geq \sqrt{\sum_{s=t_0'(t)+1}^{t} g_s(\omega_s)} - \sqrt{\sum_{s=t_0'(t)+1}^{t} \sum_{k=1}^{K} \omega_s^k c_{s-1}^k} \,.$$

$\qquad\square$

**Summary of Steps 4 to 7.** Putting together Equations (15), (17) and (18), we proved that under event $\mathcal{E}_t$, for estimates $(\tilde{\mu}_s)_{s \leq t}$ such that we have a bound $c_s^k$ on $(\tilde{\mu}_s^k - \mu^k)^2$ for all $s \in \{t_0'(t) + 1, \ldots, t\}$ and $k \in [K]$,

$$\sqrt{\inf_{\lambda \in \Lambda_m(\mu)} \|\mu - \lambda\|_{D_{W_t}}^2} + 2\sqrt{C_t} \geq \sqrt{\sum_{s=t_0'(t)+1}^{t} g_s(\omega_s)} \,.$$

Note that $\inf_{\lambda \in \Lambda_m(\mu)} \|\mu - \lambda\|_{D_{W_t}}^2 \leq t \max_{\omega \in \Delta_K} \inf_{\lambda \in \Lambda_m(\mu)} \|\mu - \lambda\|_{D_\omega}^2 = 2tH_\mu$. We then get

$$\inf_{\lambda \in \Lambda_m(\mu)} \|\mu - \lambda\|_{D_{W_t}}^2 \geq \sum_{s=t_0'(t)+1}^{t} g_s(\omega_s) - 4C_t - 4\sqrt{2tC_tH_\mu} \,,$$

which ends proving Theorem 5.

**Step** 8**. No-regret property.** The first $t_0$ rounds are used to initialize our algorithm. After that, we use a learner with small regret. We will bound the gains between $t_0$ and $t'_0(t) = \max\{t_0, \sqrt{t}\}$ by $36M^2$ (see Lemma 24). We use the regret bound of the learner (refer to Definition 2 in the main paper) to get that, for some additional low-order term $C_{\mathcal{L}}(K, B)\sqrt{t}$, and by combining Theorems 4 and 5:

$$
\begin{aligned}
2\beta_{t,\delta} &\geq \sum_{s=t'_0(t)+1}^{t} g_s(\omega_s) - h_\delta(t) - r(t) - 4C_t - 4\sqrt{2tC_tH_\mu} \\
&\geq \sum_{s=t_0+1}^{t} g_s(\omega_s) - h_\delta(t) - r(t) - 4C_t - 4\sqrt{2tC_tH_\mu} - \max\{\sqrt{t} - t_0, 0\}36M^2 \\
&\geq \max_{\omega \in \Delta_K} \sum_{s=t_0+1}^{t} g_s(\omega) - h_\delta(t) - r(t) - 4C_t - 4\sqrt{2tC_tH_\mu} - C_{\mathcal{L}}(K, B)\sqrt{t} \\
&\quad - \max\{\sqrt{t} - t_0, 0\}36M^2 \, .
\end{aligned}
$$

A specific upper bound on the regret for the learner AdaHedge used in the implementation of MISLID is mentioned in Lemma 27.

**Step** 9**. From the optimal gain to the lower bound value.** Finally, we can relate the optimal optimistic gain of the learner to the value of the lower bound. Using the optimism (Lemma 23),

$$
\max_{\omega \in \Delta_K} \sum_{s=t_0+1}^{t} g_s(\omega) \geq \max_{\omega \in \Delta_K} \sum_{s=t_0+1}^{t} \inf_{\lambda \in \Lambda_m(\mu)} \|\mu - \lambda\|_{D_\omega}^2 = (t - t_0) \underbrace{\max_{\omega \in \Delta_K} \inf_{\lambda \in \Lambda_m(\mu)} \|\mu - \lambda\|_{D_\omega}^2}_{= 2H_\mu} \, .
$$

**Step** 10**. Computing the sample complexity.** We thus have obtained an inequality of the form

$$
\begin{aligned}
2\beta_{t,\delta} &\geq 2tH_\mu - h_\delta(t) - r(t) - 4C_t - 4\sqrt{2tC_tH_\mu} - C_{\mathcal{L}}(K, B)\sqrt{t} - 2t_0H_\mu \\
&\quad - \max\{\sqrt{t} - t_0, 0\}36M^2 \, ,
\end{aligned}
$$

from which we can obtain the desired sample complexity bound. Remember that

- $\beta_{t,\delta} := \min\left\{\beta_{t,\delta}^{\mathrm{uns}}, \beta_{t,\delta}^{\mathrm{lin}}\right\}$
- $r(t) := M^2\sqrt{2t\left(\log(4 \times 5t^4) + d\log\frac{6(M+\varepsilon)Lt}{\sqrt{\Gamma(A)}} + K\log\max\{4\varepsilon t, 1\}\right)} + 2 + 8M$
- $h_\delta(t) := \sqrt{8\beta_{t,\delta}f(t)} + f(t)$
- $f(t) := 2\min\left\{\alpha_t^{\mathrm{uns}}, \alpha_t^{\mathrm{lin}} + 2t(LK + 1)^2\varepsilon^2\right\}$

where:

$$\beta_{t,\delta}^{\mathrm{uns}} := 2K\overline{W}\left(\frac{1}{2K}\log\frac{2e}{\delta} + \frac{1}{2}\log(8eK\log t)\right),$$

$$\beta_{t,\delta}^{\mathrm{lin}} := \frac{1}{2}\left(4\sqrt{t}\varepsilon + \sqrt{2}\sqrt{1 + \log\frac{1}{\delta} + \left(1 + \frac{1}{\log(1/\delta)}\right)\frac{d}{2}\log\left(1 + \frac{t}{2d}\log\frac{1}{\delta}\right)}\right)^2,$$

$$\alpha_t^{\mathrm{uns}} := 2K\overline{W}\left(\frac{1}{2K}\log(14et^3) + \frac{1}{2}\log(8eK\log t)\right) = \beta_{t,1/5t^3}^{\mathrm{uns}},$$

$$\alpha_t^{\mathrm{lin}} := \log(5t^2) + d\log\left(1 + \frac{t}{2d}\right),$$

$$c_t^k := \min\left\{8(LK+1)^2\varepsilon^2 + 4\alpha_{t^2}^{\mathrm{lin}}\|\phi_k\|_{V_t^{-1}}^2, \frac{2\alpha_{t^2}^{\mathrm{uns}}}{N_t^k}, 4M^2\right\},$$

$$C_t := \sum_{s=t_0+1}^{t}\sum_{k=1}^{K} w_s^k c_{s-1}^k \le 8(LK+1)^2\varepsilon^2 t + 2\alpha_{t^2}^{\mathrm{lin}}\left(\sqrt{2t\log(5t^2)} + d\log\left(1 + \frac{t}{d}\right)\right),$$

$$C_t \le 4M^2\sqrt{2t\log(5t^2)} + 4M^2K + 2K\alpha_{t^2}^{\mathrm{uns}}\log(et).$$

Combining this bound with Lemma 19 proves Theorem 2 in the main paper.

### F.4 Using several estimates

If we employ two sets of estimates, with corresponding optimism functions $(g_s^i(\omega))_{i\in\{1,2\}}$ and bounds $c_{i,s}^k$, we get for $i \in \{1,2\}$,

$$\inf_{\lambda\in\Lambda_m(\mu)}\|\mu - \lambda\|_{D_{W_t}}^2 \ge \max_{i\in\{1,2\}}\left(\sum_{s=t_0+1}^{t} g_s^i(\omega_s) - 4C_t^i - 4\sqrt{2tC_t^i H_\mu}\right)$$

$$\ge \sum_{s=t_0'(t)+1}^{t}\min_{i\in\{1,2\}} g_s^i(\omega_s) - \min_{i\in\{1,2\}}\left(4C_t^i + 4\sqrt{2tC_t^i H_\mu}\right),$$

where the quantity $C_t^i$ is similarly defined as $C_t$, with respect to gains $g_t^i$.

Since the minimum of concave functions is concave, $g_s : \omega \mapsto \min_{i\in\{1,2\}} g_s^i(\omega)$ is concave (which allows the use of a regret-minimizing algorithm, see Subsection F.6). It satisfies the inequality of Lemma 23 and its gradient is the gradient of $g_s^{i^\star}(\omega)$ for $i^\star \in \arg\min_{i\in\{1,2\}} g_s^i(\omega)$.

### F.5 Aggressive Optimism

If we are happy with an algorithm which is within a factor 2 of the lower bound for the $\log\frac{1}{\delta}$ term instead of insisting on a factor 1, we can use a different, more aggressive optimism. Take

$$\widehat{g}_s(\omega_s) := 2\sum_{k=1}^{K}\omega^k\left((\tilde{\mu}_{s-1}^k - \lambda_s^k)^2 + c_{s-1}^k\right) \quad \text{where } \lambda_s := \arg\min_{\lambda\in\Lambda_m(\tilde{\mu}_{s-1})}\|\tilde{\mu}_{s-1} - \lambda\|_{D_{\omega_s}}^2.$$

The main difference is that the term added to $(\tilde{\mu}_{s-1}^k - \lambda_s^k)^2$ is of order $c_{s-1}^k$ instead of $\sqrt{c_{s-1}^k}$. In an unstructured bandit, that means $1/N_t^k$ instead of $1/\sqrt{N_t^k}$. Let us prove the counterpart to Lemma 23 for these new gains:

**Lemma 26.** *For all $\omega \in \Delta_K$, $\widehat{g}_s(\omega_s) \ge \inf_{\Lambda\in\lambda_m(\mu)}\|\mu - \lambda\|_{D_\omega}^2$.*

*Proof.* For all $k \in [K]$ and $\lambda \in \mathbb{R}^K$, using Lemma 20

$$(\mu^k - \lambda^k)^2 \le 2(\tilde{\mu}_{s-1}^k - \lambda^k)^2 + 2(\mu^k - \tilde{\mu}_{s-1}^k)^2 \le 2(\tilde{\mu}_{s-1}^k - \lambda^k)^2 + 2c_{s-1}^k.$$

Then, since $\omega \in \Delta_K$

$$2 \sum_{k=1}^{K} \omega^k \left( (\tilde{\mu}_{s-1}^k - \lambda^k)^2 + c_{s-1}^k \right) \geq \sum_{k=1}^{K} \omega^k (\mu^k - \lambda_s^k)^2 = \|\mu - \lambda_s\|_{D_\omega}^2 \geq \inf_{\Lambda \in \lambda_m(\mu)} \|\mu - \lambda\|_{D_\omega}^2 .$$

$\square$

Then, using Lemma 26 and the definition of $\lambda_s$, we have

$$\hat{g}_s(\omega) - 2 \inf_{\lambda \in \Lambda_m(\tilde{\mu}_{s-1})} \|\tilde{\mu}_{s-1} - \lambda\|_{D_{\omega_s}}^2 = \sum_{k=1}^{K} \omega_s^k \left[ 2(\tilde{\mu}_{s-1}^k - \lambda_s^k)^2 + 2c_{s-1}^k - 2(\tilde{\mu}_{s-1}^k - \lambda_s^k)^2 \right]$$

$$= 2 \sum_{k=1}^{K} \omega_s^k c_{s-1}^k .$$

So now we can prove a counterpart to Step 7 in the proof of Theorem 5:

$$\sum_{s=t_0+1}^{t} \inf_{\lambda \in \Lambda_m(\tilde{\mu}_{s-1})} \|\tilde{\mu}_{s-1} - \lambda\|_{D_{\omega_s}}^2$$

$$= \frac{1}{2} \sum_{s=t_0+1}^{t} \hat{g}_s(\omega_s) - \frac{1}{2} \sum_{s=t_0+1}^{t} \left( \hat{g}_s(\omega_s) - 2 \inf_{\lambda \in \Lambda_m(\tilde{\mu}_{s-1})} \|\tilde{\mu}_{s-1} - \lambda\|_{D_{\omega_s}}^2 \right)$$

$$\geq \frac{1}{2} \sum_{s=t_0+1}^{t} \hat{g}_s(\omega_s) - C_t .$$

## F.6 Regret of AdaHedge

**Lemma 27** ([10]). *On the online learning problem with $K$ arms and gains $g_s(\omega) := \sum_{k \in [K]} \omega^k U_s^k$ for $s \in [t]$, AdaHedge, predicting $(\omega_s)_{s \in [t]}$, has regret*

$$R_t := \max_{\omega \in \Delta_K} \sum_{s=1}^{t} g_s(\omega) - g_s(\omega_s) \leq 2\sigma \sqrt{t \log(K)} + 16\sigma(2 + \log(K)/3) ,$$

*where* $\sigma := \max_{s \leq t} \left( \max_{k \in [K]} U_s^k - \min_{k \in [K]} U_s^k \right) .$

We recall here the "gradient trick", which we can use to employ AdaHedge on any concave gains. If for any time $t > 0$, the loss function $\ell_t$ at that time is convex, then for all $\omega \in \Delta_K$,

$$\sum_{s=1}^{t} \ell_t(\omega_t) - \ell_t(\omega) \leq \sum_{s=1}^{t} (\omega_t - \omega)^\top \nabla \ell_t(\omega_t)$$

Running a regret-minimizing algorithm with loss $\bar{\ell}_t(\omega) = \omega^\top \nabla \ell_t(\omega_t)$ then leads to a regret bound on $\ell_t$.

## F.7 Technical tools

**Generic bounds on vector norms.**

**Lemma 28.** *Let* $\theta \in \mathbb{R}^d, \eta \in \mathbb{R}^K$ *be such that* $\|\eta\|_\infty \leq \varepsilon$ *and* $\|A\theta + \eta\|_\infty \leq M$. *Then*

$$\|\theta\|_2 \leq \frac{M + \varepsilon}{\sqrt{\Gamma(A)}} ,$$

*where* $\Gamma(A) := \max_{\omega \in \Delta_K} \sigma_{\min} \left( \sum_{k=1}^{K} \omega^k \phi_k \phi_k^\top \right)$, *where* $\sigma_{\min}(M)$ *is the minimal singular value of matrix* $M$.

*Proof.* For $\lambda := A\theta + \eta$ with $\|\eta\|_\infty \le \varepsilon$ and $\|\lambda\|_\infty \le M$,

$$\|A\theta\|_\infty = \max_{k\in[K]} \left|\phi_k^\top \theta\right| \ge \|\theta\|_2 \min_{u\in\mathbb{R}^d:\|u\|_2=1} \max_{k\in[K]} \left|\phi_k^\top u\right| , \tag{19}$$

using that the value for $\theta/\|\theta\|_2$ is larger than the minimum over $u \in \mathbb{R}^d$ with $\|u\|_2 = 1$. On the other hand, successively using the triangle inequality and the boundedness assumptions,

$$\|A\theta\|_\infty \le \|A\theta + \eta\|_\infty + \|\eta\|_\infty \le M + \varepsilon . \tag{20}$$

Note also that

$$\min_{u\in\mathbb{R}^d:\|u\|_2=1} \max_{k\in[K]} \left|\phi_k^\top u\right|^2 = \min_{u\in\mathbb{R}^d:\|u\|_2=1} \max_{\omega\in\Delta_K} \|u\|^2_{\left(\sum_{k=1}^K \omega^k \phi_k \phi_k^\top\right)} \ge \underbrace{\max_{\omega\in\Delta_K} \sigma_{\min}\left(\sum_{k=1}^K \omega^k \phi_k \phi_k^\top\right)}_{:= \Gamma(A)} ,$$
$$\tag{21}$$

where the inequality stems from the min-max theorem (principle for singular values). Finally, by combining the three inequalities (19), (20) and (21), $\|\theta\|_2 \le \frac{M+\varepsilon}{\sqrt{\Gamma(A)}}$ .

$\square$

The term $\Gamma(A)$ depends only on the set of linear features $\{\phi_k\}_{k\in[K]}$. In the unstructured case (where $\phi_k = e_k$), we have $\Gamma(A) = \frac{1}{K}$. However, in a structured case with $d \ll K$, $\Gamma(A)$ can be much smaller. For instance, when $A$ contains the canonical basis of $\mathbb{R}^d$, we have $\Gamma(A) \ge \frac{1}{d}$.

## G Experimental evaluation

### G.1 Computational architectures

Experiments on simulated datasets (Experiments (A), (B), (C)) were run on a personal computer (processor: Intel Core i7 $-$ 8750H, cores: 12, frequency: 2.20GHz, RAM: 16GB).

Experiment (D) was run on a personal computer (processor: Intel Core i7 $-$ 9700K, cores: 8, frequency: 3.60GHz, RAM: 16GB).

Experiment (E) was run on a internal cluster (processor: Westmere E56xx/L56xx/X56xx (Nehalem$-$C), cores: 24, frequency: 3.2GHz, RAM: 155GB).

### G.2 License for the assets

**Experiment (D).** The drug repurposing dataset for epilepsy was proposed in [35], and made publicly available under the MIT license.

**Experiment (E).** The original dataset Last.fm is publicly available online at `https://www.last.fm/` under CC BY-SA 4.0.

**Experimental code.** The code hosted at `https://github.com/clreda/misspecified-top-m` is under MIT license.

### G.3 Extracting representations from real datasets

We describe in detail the procedure we adopted to extract misspecified linear representations from the real-world datasets of Experiment (D) and (E). In both cases, we adopted a very similar procedure based on training neural networks as the one used in [34]. We describe all its steps for the sake of completeness.

**Step** 1. **(Data preprocessing)** First, we start from preprocessing the raw data to obtain a dataset containing tuples of the form $(\phi, x)$, where $\phi \in \mathbb{R}^d$ is an arm feature and $x \in \mathbb{R}$ is a reward. The drug repurposing dataset used in [35] (hosted on their repository) is already available in this form, with a total of 509 arms representing different drugs, $d = 67$ features representing genes, and, for each of

them, 18 reward samples representing the responses of 18 different patients to such drugs. Out of those 509 arms, we filter out those which outcomes are unknown (associated "true" scores are set to 0, according to the file of scores available on the same repository). Then 175 arms (representing either antiepileptics, with score equal to 1, and proconvulsants, with score equal to $-1$) are left.

On the other hand, the Last.fm dataset is in a different form; it contains information about the music artists listened by each user of the system. As done in [34, Appendix F.4], we first preprocessed the data by keeping only artists listened by at least 120 users and users that listened at least to 10 different artists. We thus obtained $U = 1,322$ users and $A = 103$ artists. The result is a matrix in $\mathbb{R}^{U \times A}$ containing the number of times each user listened to each artist (which we treat as reward). We then extract user-artist features by applying low-rank Singular Value Decomposition on this matrix and keeping only the top 80 singular values. This yields $U$ $d$-dimensional user features, and $A$ $d$-dimensional artist features, where $d = 80$. The final user-artist features are the concatenation of the two, which yields a dataset with $U \times A$ tuples $(\phi, x) \in \mathbb{R}^d \times \mathbb{N}$ in our desired form.

**Step** 2. **(Neural-network training)**    For both datasets, the second step consists in training a neural network to regress from $\phi$ to $x$. First, we split the datasets randomly into $80\%$ training set and $20\%$ test set. Then, we train a neural network with two hidden layers of size 256, rectified linear unit activations, and a linear output layer of 8 neurons. We obtain an $R^2$ score on the test set of $0.92$ for the drug repurposing data, and $0.85$ for the Last.fm one.

**Step** 3. **(Extracting a linear representation)**    Finally, we extract a linear model from the trained neural network by taking, for each input $\phi \in \mathbb{R}^d$ in our data, the 8-dimensional features (i.e., activations) computed in the last layer together with the corresponding parameters. When specified, a subset of arm features is considered instead of the whole dataset. Then, in that case, we apply a lossless dimensionality reduction to make sure these features span the whole space. The reduced features are the one we feed into our learning algorithms (Experiment (D): $d = 5$, $K = 10$; Experiment (E.i): $d = 8$, $K = 103$; Experiment (E.ii): $d = 7$, $K = 50$). Moreover, we compute the maximum absolute error of this linear model in predicting the original data, and use that as a proxy for $\varepsilon$.

Note that, since the Last.fm data is in the form of user-artist features and, in our problem, we consider the artists only as arms, the representation we select for our experiments is obtained by choosing a user randomly among the available $U = 1,322$ ones.

Moreover, in Step 3, we apply a dimension reduction procedure on features to ensure the feature matrix is not ill-conditioned, at the cost of increasing the norm of misspecification $\varepsilon$. This is needed in order to reduce the length $t_0$ of the initialization sequence ; remember that in Appendix D.1 we showed that $t_0$ is upper-bounded by quantity $d\left\lceil \frac{2L^2}{\Gamma'(A)} \right\rceil$, where $\Gamma'(A) := \min\left\{\sigma_{\min}\left(\sum_{k \in \mathcal{B}} \phi_k \phi_k^\top\right) \mid \mathcal{B} \text{ barycentric spanner of } A \text{ of size } d\right\}$ crucially relies on the conditioning of $A$. How much misspecification is required to improve the conditioning of the matrix is an open question (which has also been raised in other recent works [34]). Ideally, one would want to learn a representation of the data which balances those two effects, but we leave such a method to future investigations.

### G.4    Numerical results for sample complexity

Table 2: Statistics (mean $\pm$ standard deviation rounded up to the next integer) for Experiment (A). Names are similar to those in the first two leftmost plots of Figure 1. Values are averaged across 500 iterations. LinGapE is not $\delta$-correct in the setting where $\varepsilon = 5$ (with $\delta = 0.05$).

| Sample complexity | LinGapE | MISLID |
|---|---|---|
| $\varepsilon = 0$ | $577 \pm 348$ | $890 \pm 546$ |
| $\varepsilon = 5$ | $553 \pm 536$ | $5,156 \pm 3,629$ |

Table 3: Statistics (mean $\pm$ standard deviation rounded up to the next integer) for Experiments (D) and (E). Names are similar to those in the plots of Figure 2. Values are averaged across 100 iterations. Note that LinGapE is not $\delta$-correct (with $\delta = 0.05$) in Experiment E.

| Sample complexity | LinGapE | MISLID |
|---|---|---|
| Experiment D | $21,593 \pm 8,296$ | $42,751 \pm 13,942$ |
| Experiment E | $10,907 \pm 4,474$ | $289,703 \pm 185,205$ |

### G.5  Tricks to reduce sample and computational complexity on large instances (D) and (E)

In large instances (more particularly on our real-life datasets in Section 5 in the main paper), the number of arms can be large, and the theoretically supported version of the algorithm MISLID might become too slow. Based on our experiments, we have decided to change some parts of the algorithm.

**No optimism.** As shown in the rightmost plot in Figure 1 in the main paper, empirical gains (i.e., without any optimistic bonus) actually considerably improve sample complexity.

**Restriction of the set of arms used in the sampling rule.** In order to compute the gains which are fed to the learner, MISLID needs to compute the closest alternative, which implies solving $m(K-m)$ quadratic optimization problems, one for each pair of arms $(i, j)$, with $i$ among the $m$ best arms and $j$ among the $K-m$ worse arms (as defined in Theorem 1 in the main paper). We observed that the majority of arms never realize the minimum over $(i, j)$ of the distance to the alternative, and in hindsight they could be ignored. We mimicked that behavior by only considering a subset of arms at each step. We kept $m+d$ arms in memory, consisting of the recent argmins $i, j$ for the closest alternative model, and sampled $d$ more among the $K-m$ worse arms. The resulting set of at most $m+2d$ arms is then used to compute the closest alternative. The gain in computational complexity is large when $K \gg d$, since we solve $m(m+2d)$ minimization problems instead of $m(K-m)$. We don't use that trick to compute the stopping rule, since we would not be guaranteed to preserve $\delta$-correctness.

**Geometric grid for testing the stopping rule.** Instead of checking the stopping criterion at each learning round of the algorithm, we suggest testing it on a geometric grid (that is, testing it for the first time at $t_1$, and then retest it at $\gamma t_1$, then at $\gamma^2 t_1$, etc. where $1 < \gamma \leq 1.3$ in practice), and restrict the computation of the stopping rule to a random subset of arms. In our experiments, we have actually used $\gamma = 1.2$. When using the geometric grid, we can obtain a sample complexity bound of the same form as in Theorem 2 in the main paper, except that $T_0(\delta)$ is replaced by $\gamma T_0(\delta)$.

Together, the sampling and stopping rule changes reduce the time needed to complete a run of the algorithm by a factor 29 on Experiment (D), while increasing the sample complexity by a factor 1.2 (refer to Table 4, comparing algorithmic versions named "AdaHedge" and "Default"). See the middle plot of Figure 3 for a comparison of the sample complexity.

Table 4: Statistics (mean $\pm$ standard deviation rounded up to the next integer) for Experiment (D), with different versions of MISLID. Names are similar to those in the center plot of Figure 3. Values are averaged across 100 iterations.

| Per run | AdaHedge | Greedy | Default |
|---|---|---|---|
| Average runtime (in sec.) | $69 \pm 20$ | $76 \pm 178$ | $1,993 \pm 1,311$ |
| Average sample complexity | $51,965 \pm 15,260$ | $52,108 \pm 125,230$ | $42,751 \pm 13,943$ |

We have also tested another learner which is less conservative than AdaHedge, to check if this improves sample complexity (note that we did not show any experiment using this trick in the main paper):

**Change of learner.** We replace AdaHedge by a Greedy/Follow-The-Leader learner combination for the computation of $(\omega_t, \lambda_t)$.

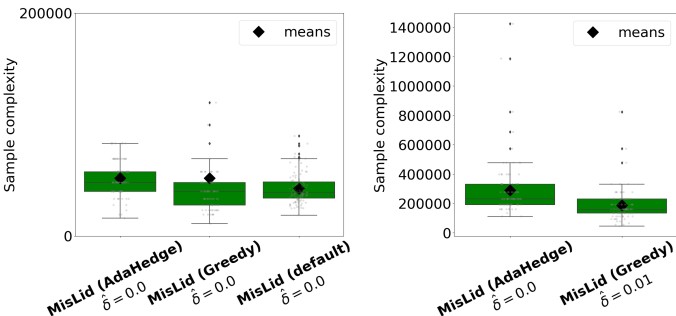

Figure 3: Comparison between default MISLID, modified MISLID using learner AdaHedge, and modified MISLID using learner Greedy (Experiment (D) *(left)*, Experiment (E)). Unfortunately, one outlier in the runs using learner Greedy in Experiment (D), above $1,200,000$ rounds, would prevent the readability of the plot if figured. To overcome this issue, we have cropped out the $y$-axis above $200,000$ in this plot.

We have run three versions of MISLID on the dataset of Experiment (D): the default MISLID, the modified version with learner AdaHedge, and another modified version with the Greedy learner. We have also launched the latter two on Experiment (E). See Figure 3.