# OpenReview forum: "Dealing With Misspecification In Fixed-Confidence Linear Top-m Identification"
_NeurIPS.cc/2021/Conference — NeurIPS 2021 Poster_

### Official Review · Reviewer_dLDo · 2021-07-14

**Rating:** 6
**Confidence:** 3

**Summary:**

This paper studied the fixed-confidence Top-m identification problem for linear bandit models. Different from previous linear bandit works, this paper considers the setting where there is some deviation from the linearity (called misspecified linear bandit models).

Assuming the upper bound of this deviation is given (denoted as eps), the misspecified linear bandit model generalizes the unstructured bandit model and the linear bandit model: when eps = 0, it recovers the linear version, and when eps -> infinity, it recovers the unstructured version.

Some interesting results have been developed under this misspecification concept, including a simple algorithm to select the top-m arms,  a lower-bound and an upper bound of the sample complexity, and they match when the maximum error rate -> 0. The authors also proved that when eps is unknown, even with the linearity assumption, we can not do anything better than what we have done for unstructured bandit models.

Finally, preliminary experimental results are provided for both real-world datasets and synthetic datasets, showing the effectiveness of the proposed algorithm.


**Limitations And Societal Impact:**


The concept of misspecification is not entirely new, and has been studied in literature as specified in the Related Work Section. Of course, misspecification hasn’t been previously explored in the pure exploration literature.

The upper bound in Thm 5 is quite complicated to interpret, not sure if it is possible to further simplify it.


My biggest concern is on the experimental results, and I am not fully convinced. Let’s take Figure 1 as an example. In the left-most figure of Figure 1, all empirical error rate is 0.0, it is possible that even if we reduce the sample complexity of LUCB, we still get an empirical error rate of 0.0. So this does not seem to be a fair comparison. In my opinion, a better way to plot the figure is that set the X-axis as the number of pulls, and Y-axis as the empirical error rates (over e.g., over 500 runs). Then you can easily compare the sample complexities between different algorithms for a certain empirical error rate by drawing a horizontal line. See e.g., Figure 4 of http://proceedings.mlr.press/v70/chen17b/chen17b.pdf


**Main Review:**


Overall I think the problems studied are quite interesting, and by introducing the concept of misspecification, the linearity assumption made in the linear bandit models is weakened, making the misspecified linear bandit better models the real-world problems.

I also like the results the paper derived,
When the upper bound of the misspecification is unknown, the problem is no easier than its unstructured version. This intuitively is true but requires non-trivial proof.
The algorithm proposed recovers the state-of-the-art dependencies in the linear case (eps = 0) and in the unstructured case (eps is large).
The algorithm has been empirically evaluated.

This paper is also well written and easy to follow when the audience is familiar with the mathematical tools/concepts being used.


**Time Spent Reviewing:**

2

---

> ### Author Response · Authors · 2021-08-06
> **Rebuttal**
>
> We thank the reviewer for their positive review and the nice words on the interest of the setting and of the theoretical results. We now address the questions raised in the review.
>
> First, the reviewer remarks that **"The upper bound in Thm 5 is quite complicated to interpret [...]"**.
>
> We acknowledge that the sample complexity upper bound is not straightforward, since it is composed of several terms arising from the linear and unstructured aspects of the problem. The full expression is available at Step $10$ in Appendix. We provided a shortened version in the main paper with the hope that it could show how the complexity depends on the constants of the problem ($d$, $K$, $\varepsilon$...). While the bound is complicated and we don't see an easy way to present it in a simpler yet rigorous form, we will rephrase the paragraph below the theorem to more distinctly list the consequences of that theorem: asymptotic optimality, no polynomial dependence in K when $\varepsilon = 0$ (linear case), transition to an optimal unstructured bound when $\varepsilon$ is large.
>
> Then, the reviewer suggests that **"[...]  it is possible that even if we reduce the sample complexity of LUCB, we still get an empirical error rate of 0.0. In my opinion, a better way to plot the figure is that set the X-axis as the number of pulls, and Y-axis as the empirical error rates [...]''**
>
> To "reduce the sample complexity of LUCB'' while still having low error rate is not something we can easily do. Indeed, the time at which the algorithm stops is random and results from the interaction between its sampling and stopping rules. It is not a parameter we can tune.
> In the fixed-confidence literature, stopping rules which guarantee $\delta$-correctness are known to be too conservative, and yield empirical error frequencies which are actually a lot lower than $\delta$. Improving that behavior is an important challenge for the design of fixed-confidence algorithms.
>
> The reviewer suggests to read from a plot the number of samples needed to reach an error rate empirically. The issue is that there is no guarantee that running the algorithm for that amount of samples gives the right answer with that error rate. That would be a fixed-budget error bound, and is not what we prove.
>
> In the paper [Chen et al. 2017] mentioned in the review, the authors write that they converted their fixed-confidence algorithms to the fixed-budget setting for the experiments. Thus, the settings of the theoretical and empirical results in that paper don't match.
>
> **References**
>
> Chen, Jiecao, et al. "Adaptive multiple-arm identification." International Conference on Machine Learning. PMLR, 2017.

---

> > ### Comment · Reviewer_dLDo · 2021-09-10
> > **still not convinced by the rebuttal on the experiment settings**
> >
> > > The issue is that there is no guarantee that running the algorithm for that amount of samples gives the right answer with that error rate
> >
> > That's why in [Chen et al. 2017] they run the experiments many times (e.g., 100 times) and report the empirical probability of the error being lower than a pre-fixed threshold.
> >
> > > Thus, the settings of the theoretical and empirical results in that paper don't match.
> >
> > That's okay, at least Chen et al. 2017 gave a fair comparison to show which algorithm requires less #samples to achieve pre-fixed error with certain confidence, while in your approach, it is unclear which algorithm is empirically better.

---

> > > ### Author Response · Authors · 2021-09-10
> > > **Reply to the follow-up**
> > >
> > > We thank the reviewer for the follow-up. We would like to provide some additional arguments on why we believe our experimental protocol to be fair.
> > >
> > > In the fixed-confidence setting, an algorithm is described by two main components, a stopping rule and a sampling rule. Both play a fundamental role in characterizing the final sample complexity: the sampling rule ensures that the algorithm collects enough information to identify the unknown correct answer (equivalently, to reduce the error of misidentification), while the stopping rule makes sure that the returned answer is correct with high probability. The experimental protocol the reviewer is suggesting is a sound technique for fairly comparing the *sampling rules* of different algorithms, but would completely ignore their *stopping rules*. In fact, once we fix a budget of samples beforehand, the stopping rule is no longer used and the resulting algorithms can be characterized by their sampling strategies alone. Therefore, such a comparison would indeed provide insights on which sampling strategy is better at reducing the empirical error rate, but it would not evaluate the actual sample complexity (i.e., stopping time) of the algorithms. Moreover, knowing which algorithm requires fewer samples to achieve a pre-defined error threshold in a certain problem does not necessarily indicate the best algorithm to run for such problem: since we do not know when to stop, and thus algorithms need to do that adaptively, we have no guarantee that the fastest algorithm to achieve the desired error threshold will also be the first to stop (and thus the one with minimal sample complexity).
> > >
> > > On the other hand, we believe that our evaluation protocol fairly compares all the considered algorithms according to their exact definition (i.e., the combination of their sampling and stopping rules) and in terms of the quantity of interest in the fixed-confidence setting (i.e., their stopping time). This matches precisely the setting we consider in our theoretical results.

---

### Official Review · Reviewer_vF9d · 2021-07-16

**Rating:** 7
**Confidence:** 3

**Summary:**

In this work the authors tackle the problem of finding the (potentially not unique) top-m arms in the misspecified linear bandit setting. An instance dependent information theoretic lower bound is given for any delta-correct algorithm. From the computationally efficient reformulation of this lower bound the authors provide, an algorithm is proposed, for which finite sample guarantees are provided. Optimality is shown in the high confidence regime. Synthetic and real-world experiments validate the statistical and computational efficiency of this scheme.

**Limitations And Societal Impact:**

There are no negative societal impacts in this theoretical paper that focuses on the top-m identification setting in misspecified linear bandits.

**Main Review:**

1. Overall, the experimental evaluation of this scheme seems interesting and well done, particularly experiments (D) and (E) on real datasets. The numerical results, especially for the synthetic experiments, are hard to parse however; aside from table 2, the results are only displayed visually. For example, the inclusion of LUCB in experiment (A) makes it difficult to compare LinGapE and MisLid. It seems as though on all synthetic simulated instances, LinGapE outperforms MisLid, potentially quite significantly. If epsilon is taken to be larger in Experiment (A) (say epsilon = 5) will we see a similar hit in accuracy as in Experiment (E)? Additional justification for the heuristic stopping rule in line 291 would be helpful. Line 317 Experiment (B), no unstructured bandit baseline algorithm appears to be shown. Line 132, is it clear that for small m (e.g. m=3 or 4 as in experiments) the combinatorial optimization over (K choose m) sets is less efficient than the proposed scheme?
2. The lower bound appears to show that for a given class of models (known epsilon,M) adaptation to the true infinity norm of eta is not possible (which is surprising). Is there some alternate assumption under which this is possible, for example knowledge on the relationship between the norms of mu and eta?
3. I wasn’t able to find where t_0 the initialization time is bounded, which seems as though it could be poor (but constant) for ill-conditioned phi.

Minor errors:
- Line 17 K is a set
- Line 36 mean rewards are inner products, not linear combinations
- Line 96 \ell_2
- Algorithm 1 updating learner with \omega_t
- Line 186 D_{N_t} should be defined with respect to an arbitrary vector, not just N_t
- Line 257: T_0(\delta) is ‘a’ solution?
- Fig 1: Experiment (B) ‘middle’
- Line 302: most -> more
- Line 310: a user-provided
- Line 361: at each time -> in each round
- Line 619 remainder
- Line 630 V_{t_0}

**Time Spent Reviewing:**

3.5

---

> ### Author Response · Authors · 2021-08-06
> **Rebuttal**
>
> We would like to thank the reviewer for their thorough feedback. We will now address their remarks.
>
> The reviewer asks **"[...] It seems as though on all synthetic simulated instances, LinGapE outperforms MisLid, potentially quite significantly. If epsilon is taken to be larger in Experiment (A) (say epsilon = 5) will we see a similar hit in accuracy as in Experiment (E)?''**
>
> One can perform this experiment by modifying line in script ExperimentA.sh in the supplemental code from "c\_values=(0 2)'' to "c\_values=5''. Indeed, as predicted by the paragraph (A) in the experimental section, having $\varepsilon$ a lot larger than $\Delta$ changes the answer set. While LUCB (with an average sample complexity larger than $60,000$) and MisLid (with an average sample complexity lower than $20,000$) preserve $\delta$-correctness with empirical error $\hat{\delta}=0$, LinGapE yields $\hat{\delta}=0.946$ (all error rates rounded up to $5$ decimal places). We plan on adding the corresponding figure to the final version of the paper to illustrate our point and thank the reviewer for that suggestion.
>
> Then, the reviewer adds the following remarks:
>
> **"Additional justification for the heuristic stopping rule in line $291$ would be helpful."**
>
> In the fixed-confidence literature, stopping thresholds which guarantee $\delta-$correctness are known to be too conservative, and yield empirical error frequencies which are actually much lower than $\delta$. This is the reason why, for the sake of fairness, as other papers did [Kaufmann and Kalyanakrishnan, 2013; Réda et al., 2021], we actually considered for all algorithms a heuristic threshold which allows decreasing the sample complexity at the risk of violating $\delta$-correctness. In practice, $\delta$-correctness is seldom violated with this stopping rule, as illustrated by our experiments. Moreover, this stopping threshold is not the reason for failure in LinGapE, as running Experiment (A) with $\epsilon=5$ and the theoretically-supported threshold for linear bandits proven by [Abbasi-Yadkori et al., 2011] still yields an error rate of $1$. One can reproduce this experiment by modifying line in script ExperimentA.sh in the supplemental code from "c\_values=(0 2)" to "c\_values=5", and "beta=heuristic" to "beta=linear". We will add this comment in the final version of the paper.
>
> **"Line 317 Experiment (B), no unstructured bandit baseline algorithm appears to be shown.''**
>
> Experiment (B) aims at observing the behaviour of MisLid under misspecification of $\epsilon$, in order to illustrate Lemma $2$, which is why no other bandit algorithms are included in the plot.
>
> **"Line 132, is it clear that for small m (e.g. $m=3$ or $4$ as in experiments) the combinatorial optimization over (K choose m) sets is less efficient than the proposed scheme?''**
>
> We verified empirically that $\left(^K_m \right)$ is greater than $(K-m) \times m$ for all values smaller than 200, and thus the minimization step we use requires solving less subproblems than the initial combinatorial optimization proposed in the lower bound from the direct application of Theorem 1 in [Kaufmann et al., 2016].
>
> **"The lower bound appears to show that for a given class of models (known epsilon,M) adaptation to the true infinity norm of eta is not possible (which is surprising). Is there some alternate assumption under which this is possible, for example knowledge on the relationship between the norms of mu and eta?''**
>
> We thank the reviewer for this interesting question. In general, any assumption that restricts the set of possible mean vectors in the model will lead to a lower complexity. The issue with the assumption "misspecified, but with unknown misspecification" is that all mean vectors verify it. The knowledge of a relationship between the norms of $\mu$ and $\eta$ could in principle be used to obtain faster algorithms. Finding practically relevant assumptions that allow adaptation would indeed be an interesting direction for future work.
>
> **"I wasn’t able to find where $t_0$ the initialization time is bounded, which seems as though it could be poor (but constant) for ill-conditioned $\phi$.''**
>
> Note that we require a sequence of $t_0$ arms such that $\sigma_{min}(\sum_{t=1}^{t_0}\phi_{k_t}\phi_{k_t}^T) \geq 2L^2$. Equivalently, $t_0\sigma_{min}(\sum_{k=1}^K \frac{N_{t_0}^k}{t_0} \phi_{k}\phi_{k}^T) \geq 2L^2$. This implies that $t_0 \geq \frac{2L^2}{\Gamma(A)}$ where $\Gamma(A)$ is the quantity defined in Lemma 26 and indeed depends on the conditioning of the features. Since this is a lower bound to $t_0$, the reviewer is right that $t_0$ is large if $\phi$ are ill-conditioned.
>
> Ideally, we could choose the first $t_0$ arms from the design that maximizes the minimum eigenvalue of the resulting design matrix (the one that appears in the definition of $\Gamma(A)$). In such case we would have that, roughly, $t_0 \leq \frac{2L^2}{\Gamma(A)}$ (up to a small additive term due to the fact that the allocation is integer valued), so such scaling is tight.
>
> In our implementation of MisLid, as described in Subsection 4.1, we do not use this approach but rather compute a barycentric spanner of the arm feature vectors. which is more common and can be done more efficiently. Then, we sample from it in a round robin fashion until the desired condition is met.
>
> Finally, we note that in Experiments (D) and (E) we apply a dimension reduction procedure on features to ensure the feature matrix is not ill-conditioned (as described in Appendix F.3), at the cost of increasing the norm of misspecification. How much misspecification is required to improve the conditioning of the matrix is an open question (which has also been raised in other recent works, see e.g. [Papini et al. 2021]). Ideally, one would want to learn a representation of the data which balances those two effects, but we leave such a method to future investigations.
>
> **References**
>
> Kaufmann, Emilie, and Shivaram Kalyanakrishnan. "Information complexity in bandit subset selection." Conference on Learning Theory. PMLR, 2013.
>
> Réda, Clémence, Kaufmann, Emilie, and Delahaye-Duriez, Andrée. "Top-m identification for linear bandits." International Conference on Artificial Intelligence and Statistics. PMLR, 2021.
>
> Abbasi-Yadkori, Yasin, Pál, Dávid, and Szepesvári, Csaba. "Improved algorithms for linear stochastic bandits." Advances in neural information processing systems 24 (2011): 2312-2320.
>
> Papini, Matteo, et al. "Leveraging good representations in linear contextual bandits.". ICML 2021.

---

> > ### Comment · Reviewer_vF9d · 2021-08-15
> > **Follow-up**
> >
> > Thank you for the cogent rebuttal. Some quick follow-ups regarding your comments:
> >
> > Experiments: plotting the y-axis in linear scale for Experiment A makes it difficult to compare the performance between LinGapE and MisLid. It would help with the overall readability of the empirical results if the authors were able to provide either tabulated results or plots with the y-axis in log scale.
> >
> > Line 132: it is still not clear to me whether the combinatorial optimization problems that arise in [Kaufmann et al., 2016] are easier to solve (even if slightly larger in number for small m) than the $(K-m)\times m$ subproblems that are newly posed. It appears as though this is not the case, as the minimization being performed here is over half-spaces which should be quite efficient, but a more precise discussion on this point would be helpful.
> >
> > $t_0$: I think it would help make this work more clear and self-contained if an explicit upper bound on $t_0$ were provided.

---

> > > ### Author Response · Authors · 2021-08-17
> > > **Reply to the follow-up**
> > >
> > > Thanks for the follow-up.
> > >
> > > We will take into account the recommended changes in the plotting of sample complexity and the upper bound on $t_0$ in the final version of the paper.
> > >
> > > We also will develop more explicitly the difference between our approach and the "naive" combinatorial one for the computation of the alternative model in the lower bound. In particular, we will stress upon the fact that our decomposition uses half-spaces, which makes the optimization as simple as it can be; whereas, in the original form of the lower bound, the shape of the sets in which we can decompose the alternative set is more complicated.

---

### Official Review · Reviewer_Q9sG · 2021-07-16

**Rating:** 7
**Confidence:** 1

**Summary:**

The authors study the top-m identification for misspecified linear multi armed bandits. They propose theoretical results in the form of a lower bound on the sample complexity for any approximate algorithm for general top-m identification problem and also show that knowing the problem is misspecified without knowing the scale of the misspecification is the same as not knowing anything about the structure of the problem. They then propose the first algorithm MisLid for fixed confidence top-m identification in misspecified bandits, prove guarantees for delta-correctness and sample complexity, and perform experimental evaluation of the algorithm.


**Limitations And Societal Impact:**

The authors discuss the main limitation of their work being the computational complexity, but also present ideas on overcoming the limitations. They also briefly mention that the algorithm relies on features provided in the input data and could be subjected to bias and lack of fairness in the recommendation.

**Main Review:**

The results presented appear to be novel and practical. While the result is well presented and seems to be correct, I am unable to accurately review the paper in the context of originality and significance because I feel like I lack the domain knowledge to accurately do so.


**Time Spent Reviewing:**

5-6

---

> ### Author Response · Authors · 2021-08-06
> **Rebuttal**
>
> We thank the reviewer for their positive feedback and time reviewing the manuscript.

---

### Decision · Program_Chairs · 2021-09-28

**Decision:**

Accept (Poster)

**Comment:**

The review team seems aligned that the paper meets the NeurIPS threshold with one borderline vote. I wanted to thank the reviewers and authors for the detailed back and forth during the discussion period. I have also been informed from that process on how the authors consider their contributions and the limitations of the model and results. While several points remain hanging and there is some disagreement over the numerical results, I feel the paper has a good story to tell, and while the issue of misspecification is not new in the MAB literature, the pure exploration problem does raise its own challenges and hence this is a worthwhile paper that will have an audience. I have to agree with reviewer 2 (vF9d) about the bound on t_0. The relation to Kauffman (2016) in terms of lower bounds should also be better discussed and perhaps more broadly and more clearly laying out what additional complexity in analysis stems from misspecification relative to earlier work that study the fixed confidence setting.

**Consistency Experiment:**

NeurIPS has a long history of experimentation. In 2014, NeurIPS ran an experiment in which 10% of submissions were reviewed by two independent committees to quantify the randomness in the review process. This year, we repeated a variant of this experiment to see how the quality of the review process has changed over time.  This paper was part of the experiment and was therefore assigned to two committees (consisting of reviewers, an Area Chair, and a Senior Area Chair) that reached independent decisions.  If both committees made the same recommendation, this recommendation was followed. If a single committee recommended acceptance, the paper was accepted (with the exception of a few cases in which the other committee identified what we considered a fatal flaw, e.g., an error in a key result).

This copy’s committee reached the following decision: **Accept (Poster)**

The other committee assigned to the paper recommended **Reject**.  You can find the other set of reviews, along with any follow up discussion with the authors here:
https://openreview.net/forum?id=ALg4gT0DL4Axd